# miR-1 coordinately regulates lysosomal v-ATPase and biogenesis to impact proteotoxicity and muscle function during aging

Isabelle Schiffer[1†], Birgit Gerisch[1†], Kazuto Kawamura[1†], Raymond Laboy[1], Jennifer Hewitt[1,2], Martin Sebastian Denzel[1,3], Marcelo A Mori[4,5,6], Siva Vanapalli[2], Yidong Shen[7], Orsolya Symmons[1], Adam Antebi[1,3]*

[1]Max Planck Institute for Biology of Ageing, Cologne, Germany; [2]Department of Chemical Engineering, Texas Tech University, Lubbock, United States; [3]Cologne Excellence Cluster on Cellular Stress Responses in Aging-Associated Diseases (CECAD), University of Cologne, Cologne, Germany; [4]Laboratory of Aging Biology, Department of Biochemistry and Tissue Biology, University of Campinas (UNICAMP), Campinas, Brazil; [5]Experimental Medicine Research Cluster (EMRC), University of Campinas (UNICAMP), Campinas, Brazil; [6]Obesity and Comorbidities Research Center (OCRC), University of Campinas (UNICAMP), Campinas, Brazil; [7]State Key Laboratory of Cell Biology, Innovation Center for Cell Signaling Network, CAS Center for Excellence in Molecular Cell Science, Shanghai Institute of Biochemistry and Cell Biology, Chinese Academy of Sciences, University of Chinese Academy of Sciences, Shanghai, China

*For correspondence:
antebi@age.mpg.de

[†]These authors contributed equally to this work

Competing interests: The authors declare that no competing interests exist.

**Abstract** Muscle function relies on the precise architecture of dynamic contractile elements, which must be fine-tuned to maintain motility throughout life. Muscle is also plastic, and remodeled in response to stress, growth, neural and metabolic inputs. The conserved muscle-enriched microRNA, miR-1, regulates distinct aspects of muscle development, but whether it plays a role during aging is unknown. Here we investigated *Caenorhabditis elegans* miR-1 in muscle function in response to proteostatic stress. *mir-1* deletion improved mid-life muscle motility, pharyngeal pumping, and organismal longevity upon polyQ35 proteotoxic challenge. We identified multiple vacuolar ATPase subunits as subject to miR-1 control, and the regulatory subunit *vha-13*/ATP6V1A as a direct target downregulated via its 3′UTR to mediate miR-1 physiology. miR-1 further regulates nuclear localization of lysosomal biogenesis factor HLH-30/TFEB and lysosomal acidification. Our studies reveal that miR-1 coordinately regulates lysosomal v-ATPase and biogenesis to impact muscle function and health during aging.

## Introduction

Aging is the major cause of progressive decline in all organ systems throughout the body. This is particularly evident in the musculature and often manifests as sarcopenia, the loss of muscle mass and strength. In fact, muscle frailty is a hallmark of tissue aging seen in species as diverse as worms, flies, mice, and humans (*Cruz-Jentoft et al., 2010*; *Demontis et al., 2013*; *Herndon et al., 2002*; *Martinez et al., 2007*; *Miller et al., 2008*; *Nair, 2005*). At the molecular level, frailty is often accompanied by a decline in muscle structure and function, as well as alterations in muscle proteostasis and metabolism. Nevertheless, muscle can respond positively to exercise and stress and rejuvenate

even into older age, showing remarkable plasticity (*Cartee et al., 2016*; *Distefano and Goodpaster, 2018*; *Pollock et al., 2018*). A molecular study of muscle aging and plasticity in a genetically tractable model could shed light on the fundamental aspects of these processes.

microRNAs are small 22–26 nucleotide RNAs that bind with complementarity through their seed sequence to target mRNAs to downregulate gene expression (*Gu and Kay, 2010*). They can work as molecular switches or fine-tune gene regulation, and typically have multiple targets, thereby coordinating cellular programs. Many microRNAs are expressed in a tissue-specific manner and regulate programs cell autonomously, and could therefore potentially serve as tissue-specific therapeutic targets (*Guo et al., 2015*; *Panwar et al., 2017*). In addition, some microRNAs are found circulating in serum (*Chen et al., 2008*; *Weber et al., 2010*), raising the possibility that they can act cell non-autonomously as well.

miR-1 homologs are muscle-enriched microRNAs that are highly conserved in evolution and important for mammalian heart and muscle development (*Zhao et al., 2007*). Like in other animals, *Caenorhabditis elegans mir-1* is expressed in body wall ('skeletal') and pharyngeal ('cardiac') muscle (*Simon et al., 2008*). Deletion mutants are viable and exhibit modest changes in the behavior of the neuromuscular junction (*Simon et al., 2008*) and autophagy (*Nehammer et al., 2019*), but little else is known of its normative function. In this work, we examined the potential role of miR-1 in regulating muscle function and proteostasis. Surprisingly, we found that under polyQ35 proteotoxic challenge, *mir-1* mutation prevents aggregate formation and increases organismal motility and pharyngeal contractility during aging. Mechanistically, miR-1 regulates expression of several lysosomal v-ATPase subunits and targets a crucial regulatory component, *vha-13*/ATP6V1A, via its 3'UTR to impact proteotoxicity and longevity. Further, miR-1 regulates lysosomal acidification and nuclear localization of HLH-30/TFEB, a key regulator of lysosome biogenesis. These studies reveal an unexpected role of miR-1 in coordinating lysosomal function and health during aging.

## Results

### *mir-1* mutation improves muscle motility upon polyQ35 challenge

*mir-1* is a highly conserved microRNA expressed predominantly in *C. elegans* muscle tissues including pharyngeal muscle, body wall muscle, and sex muscles (*Andachi and Kohara, 2016*; *Martinez et al., 2008*). While microRNAs are known as important developmental regulators (*Ivey and Srivastava, 2015*), miR-1 expression also persists throughout adulthood, declining by half from day 1 to day 14 of adulthood (*Figure 1—figure supplement 1A*, data from *Zhou et al., 2019*). To unravel molecular miR-1 function, we first characterized the nature of several *mir-1* mutations. The reference alleles (*n4101, n4102*) consist of large deletions of the region that could have confounding effects on nearby loci. We therefore focused on *gk276*, a smaller 192 base pair deletion that removes the *mir-1* coding region as well as the downstream non-coding region, and used this deletion in our experiments. We also generated an independent *mir-1* allele, *dh1111*, by CRISPR genome engineering (*Figure 1A*), which deletes a 53 base pair region within the *mir-1* locus. Both alleles failed to express the mature miRNA as measured by TaqMan qPCR (*Figure 1—figure supplement 1B*), and thus are *mir-1* null mutants.

To investigate *mir-1* function, we measured several physiological parameters. We found that *mir-1* mutants *gk276* and *dh1111* developed normally, had normal brood size, and near normal life span (*Figure 1—figure supplement 1C–E*, *Supplementary file 1a*). Additionally, motility on solid media, thrashing in liquid medium, and pharyngeal pumping ability determined at days 8 and 14 of adulthood were similar to N2 wild-type (WT) controls (*Figure 1—figure supplement 1F, G*, *Supplementary file 1b*). Since *mir-1* deletion showed little obvious phenotype on its own, we reasoned that some physiological differences might emerge under stress conditions, hence we tested heat and proteotoxic stress. We saw little distinction from WT with moderate heat stress by growth at 25°C nor heat shock resistance at 35°C (*Figure 1—figure supplement 1E, H*, *Supplementary file 1a*).

We next tested the idea that *mir-1* mutants might differ in their response to proteotoxic stress. To do so, we used a strain expressing polyglutamine-35 tracts fused to yellow fluorescent protein (YFP) under the control of the muscle myosin-specific *unc-54* promoter (*unc-54p::Q35::YFP*), which has been used previously in models of proteotoxicity and Huntington's disease (*Brignull et al.,*

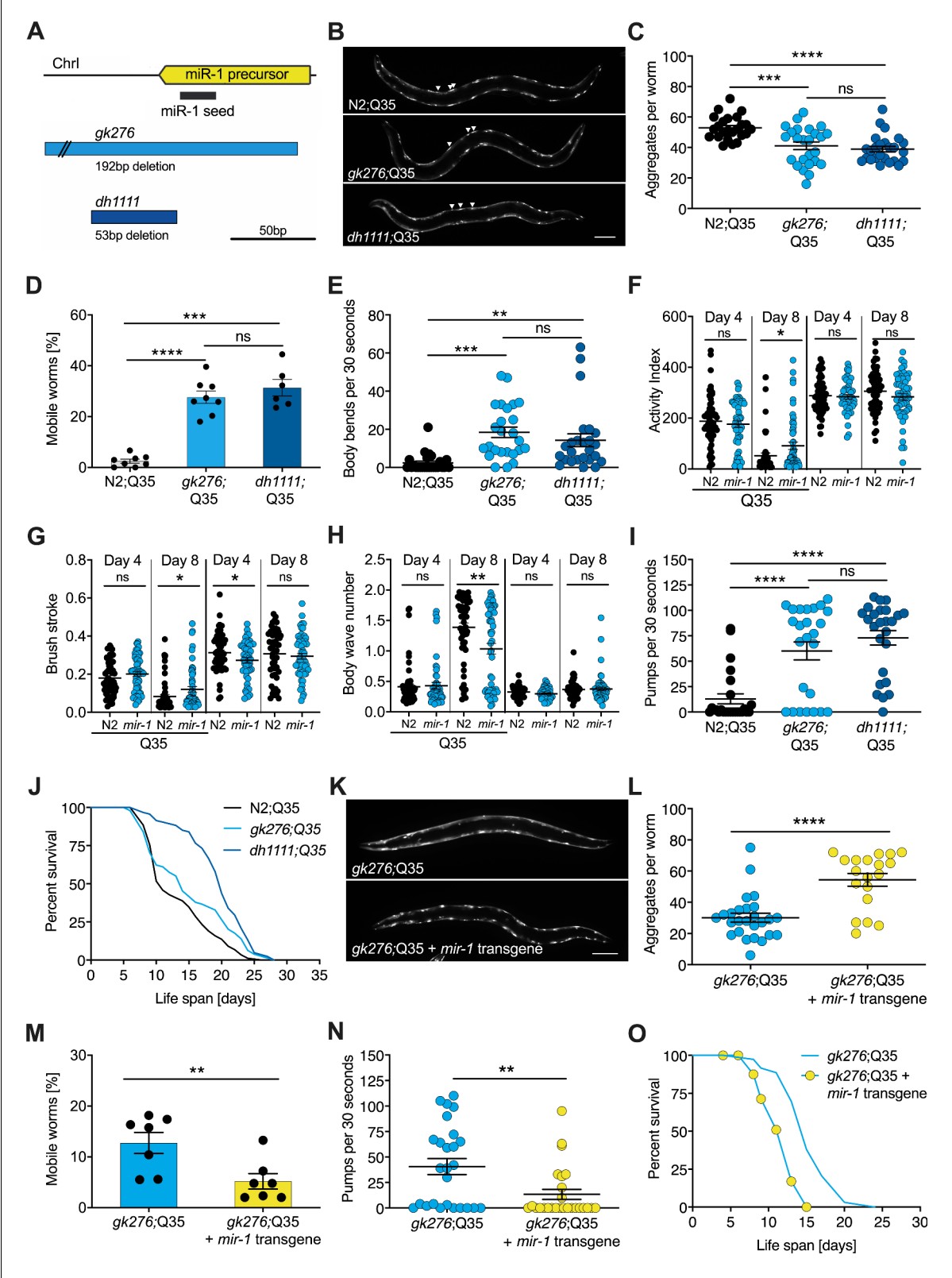

**Figure 1.** *mir-1* mutants exhibit improved motility upon polyQ35 challenge. (**A**) Schematic showing the *mir-1* locus, deletion alleles *mir-1(gk276)* and *mir-1(dh1111)* (see Materials and methods). (**B**) Representative images of N2 wild-type (WT) and *mir-1* mutant animals expressing *unc-54p::Q35::YFP* (Q35) showing loss of aggregates in the *mir-1* background on day 4 of adulthood. Arrowheads point to aggregates. Scale bar 100 μm. (**C**) Quantification of Q35::YFP aggregates in (**B**). Each dot represents one animal, mean ± SEM from one representative experiment, N = 3, one-way

*Figure 1 continued on next page*

*Figure 1 continued*

ANOVA, Tukey's multiple comparisons test, *** p<0.001, ****p<0.0001, ns, not significant. (D) Motility of indicated *mir-1* alleles and N2 (WT) animals expressing *unc-54p::Q35::YFP*, measured by the circle test, day 8 of adulthood. Percent worms that left the circle after 30 min. Each dot represents one experiment. Mean ± SEM of N = 6–8. One-way ANOVA, ***p<0.001, ****p<0.0001, ns, not significant. (E) Motility of *mir-1* alleles and N2 animals expressing *unc-54p::Q35::YFP*, measured by the thrashing assay. 25 worms per condition, each dot represents one animal, mean ± SEM from one representative experiment, N = 4, one-way ANOVA, **p<0.01 ***p<0.001, ns, not significant. CeleST analysis of activity index (F), brush stroke (G), and body wave number (H) comparing N2 and *mir-1(gk276)* mutant animals with or without *unc-54p::Q35::YFP* at days 4 and 8 of adulthood. Each dot represents one animal. t-test, *p<0.05 **p<0.01, ns, not significant. (I) Pharyngeal pumping rate measured on day 8 of adulthood in *mir-1* alleles and WT animals expressing *unc-54p::Q35::YFP*. Each dot represents one animal, mean ± SEM from one representative experiment, N = 3, one-way ANOVA, ****p<0.0001, ns, not significant. (J) Life span experiments of *mir-1* mutants and N2 animals expressing *unc-54P::Q35::YFP*. One experiment of N = 3. Log-rank test: N2;Q35 vs. *gk276*;Q35: p=0.03. N2;Q35 vs. *dh1111*;Q35: p<0.0001. (K) Presence of an extrachromosomal *mir-1* transgene brings back aggregates in *mir-1(gk276)* mutants at day 4 of adulthood, showing rescue by the transgene. Transgenic worms were compared to non-transgenic segregants of the same strain. Scale bar 100 μm. (L) Quantification of *Q35::YFP* aggregates (from K), each dot represents total aggregate number of one animal, mean ± SEM from one representative experiment, N = 3, t-test, ****p<0.0001. (M) Motility of *mir-1(gk276)*;Q35 animals in the presence or absence of the *mir-1* transgene at day 8 of adulthood, measured by circle test. Transgenic worms were compared to non-transgenic segregants of the same strain. Each dot represents one experiment, mean ± SEM of N = 7. t-test, **p<0.01. (N) Pharyngeal pumping rate of *mir-1(gk276)*; Q35 animals in the presence or absence of the *mir-1* transgene. Transgenic worms were compared to non-transgenic segregants of the same strain. Each dot represents one animal, mean ± SEM from one representative experiment, N = 7, t-test, **p<0.01. (O) Life span experiments of *mir-1(gk276)*;Q35 animals in the presence or absence of the *mir-1* transgene. Transgenic worms were compared to non-transgenic segregants of the same strain. One representative experiment of N = 5, log-rank test, p<0.0001.

The online version of this article includes the following figure supplement(s) for figure 1:

**Figure supplement 1.** Characterization of *mir-1* mutant physiology under normal conditions and proteotoxic models.

*2006*; *Morley et al., 2002*). In these strains, initially soluble proteins become sequestered into insoluble aggregates, visible as foci in the muscle of the worm. Worms expressing the polyQ35 stretches showed progressive age-dependent protein aggregation, which became clearly visible by day 4 of adulthood. Interestingly, we observed that *mir-1* mutants displayed significantly less aggregates (avg. 38 and 36, day 4 of adulthood) compared to age-matched WT controls (avg. 53) at this time (*Figure 1B, C*, *Supplementary file 1b*).

We further investigated the possible effect of proteotoxicity by measuring organismal motility of Q35 strains using the circle test. This test measures how many worms crawl out of a 1 cm circle within 30 min on day 8 of adulthood on agar plates. Day 8 was chosen as a timepoint where paralysis becomes visible, but animals are still viable. We observed that about 30% of the *mir-1*;Q35 mutants left the circle while only 2.5% of the control Q35 worms did so (*Figure 1D*, *Supplementary file 1b*). We also measured motility by counting the thrashing rate in liquid medium on day 8 of adulthood. Though the degree of paralysis was more variable, both *mir-1* mutants *gk276* and *dh1111* were significantly more mobile (avg. 16, 15 body bends) compared to age-matched Q35 controls (avg. two body bends, *Figure 1E*, *Supplementary file 1b*).

To further characterize miR-1's impact on muscle function, we performed CeleST analysis (*Restif et al., 2014*), a MATLAB-based algorithm that extracts various parameters of movement (wave initiation, travel speed, brush stroke, activity index) and gait (wave number, asymmetry, curl, stretch) from video footage. Typically activity measures decline with age, whereas gait abnormalities increase with age (*Restif et al., 2014*). We scored animals focusing on days 4 and 8 of adulthood comparing *mir-1(gk276)* and WT in the presence and absence of the Q35 array. In the absence of Q35, *mir-1* mutants performed similar to WT (e.g., body wave) or were slightly different (wave initiation, travel speed, stretch) for some parameters (*Supplementary file 1c*). In the presence of Q35, however, *mir-1(gk276)* mutants did significantly better than WT for several measures of activity (activity index, brush stroke, travel speed, wave initiation rate) and gait (number of body waves) on day 8 (*Figure 1F–H*, *Figure 1—figure supplement 1I–M*, *Supplementary file 1c*), consistent with the idea that *mir-1* mutation protects against Q35 proteotoxic stress during mid-life. *mir-1* is also highly expressed in the pharynx, a contractile tissue similar to cardiac muscle. We observed that the pumping rate of *mir-1*;Q35 worms was significantly increased (*gk276*, avg. 60; *dh1111*, avg. 78) compared to Q35 controls (avg. 16) on day 8 of adulthood (*Figure 1L*, *Supplementary file 1b*). Furthermore, we found that the life span of *mir-1(dh1111)*;Q35 strains was significantly extended (n = 3, mean 13 [WT] vs. 19 [*dh1111*] days) compared to Q35 alone (*Figure 1J*, *Supplementary file 1a*). A similar trend was seen with *gk276* in two out of three experiments (*Figure 1J*, *Supplementary file*

*1a*). These findings suggest that *mir-1* reduction can counter the intrinsic and systemic detrimental effects of Q35 proteotoxic challenge.

Introducing a WT *mir-1* transgene into the *mir-1(gk276)* background restored normal levels of mature miR-1 (*Figure 1—figure supplement 1N*) and largely reversed *mir-1* phenotypes, showing increased Q35 aggregates, decreased motility, pumping rate, and life span compared to non-transgenic controls (*Figure 1K–O*, *Supplementary file 1a, b*). The similarity of behavior among the different *mir-1* alleles and the rescue of these phenotypes by the WT *mir-1* transgene demonstrates that the *mir-1* mutation is causal for removing muscle aggregates and improving mid-life motility.

To determine whether *mir-1* loss generally protects against proteotoxic stress, we examined two other muscle-specific proteotoxic models: polyQ40 and human α-synuclein (α-syn). In contrast to polyQ35, which forms aggregates during adulthood, polyQ40 aggregates were already visible during development. We found aggregate numbers significantly reduced in *mir-1(gk276)*;Q40 L4 worms (*Supplementary file 1b*). However, the motility of *mir-1(gk276)*;Q40 worms, determined by 1 min thrashing assays, was unaffected at day 3 of adulthood and reduced in two out of three experiments at day 7 of adulthood compared to Q40 controls (*Supplementary file 1b*). Similarly, aggregate numbers were reduced in *mir-1(gk276)*;α-syn worms at day 1 of adulthood compared to the α-syn control (*Supplementary file 1b*). Thrashing of *mir-1(gk276)*;α-syn worms was significantly reduced on day 1 of adulthood, but by day 5 there was no significant difference in motility between *mir-1 (gk276)*;α-syn and α-syn worms (*Figure 1—figure supplement 1O*, *Supplementary file 1b*). Taken together, *mir-1* loss reduced aggregate numbers in three different proteotoxic stress paradigms. However, reduced number of aggregates did not always directly correlate with improved motility, reinforcing the notion that protein aggregation may promote or aggravate health depending on the proteotoxic species and age (*Cohen et al., 2006*).

## v-ATPase subunits are downstream mediators of miR-1-induced motility improvement

To identify downstream mediators of protein quality control improvement in *mir-1* mutants, we used computational and proteomic approaches to find potential miR-1 targets. For the computational approach, we used publicly available prediction tools to identify genes harboring miR-1 seed binding sites in their 3′UTR, namely microRNA.org, TargetScanWorm, and PicTar (*Betel et al., 2008*; *Jan et al., 2011*; *Lall et al., 2006*). These prediction tools use distinct algorithms that often yield different and extensive sets of candidates (*Figure 2A*). Therefore, we limited ourselves mostly to candidates that were predicted as targets by all three algorithms. This yielded 68 overlapping candidates (*Figure 2A*; *Supplementary file 1d*). Interestingly, network analysis of these 68 genes using STRING (string-db.org) (*Snel et al., 2000*) revealed a tight cluster of predicted targets encoding 11/21 subunits of the vacuolar ATPase (*Figure 2B*). Five additional vacuolar ATPase genes were identified by two of the three prediction algorithms (*Figure 2—figure supplement 1A*). In addition, two genes that code transcription factors implicated in lysosome biogenesis, *daf-16*/FOXO and *hlh-30*/TFEB (*Sardiello et al., 2009*; *Settembre et al., 2011*), also contained predicted miR-1 binding sites in their 3′UTRs.

Next, we carried out a functional screen of the selected candidates. We reasoned that *mir-1* deletion likely results in an upregulation of these proteins, and that RNAi knockdown of the relevant genes should therefore abrogate the improved motility of *mir-1*;Q35 worms. Specifically, we tested 44 available RNAi clones (*Supplementary file 1e*) and scored motility using a rapid version of the circle test, measuring the ability of *mir-1*;Q35 worms grown on candidate RNAis to migrate out of a 1 cm diameter circle within 1 min (*Figure 2C*). 12/20 *mir-1*;Q35 worms left the circle when grown on luciferase control RNAi (*luci*). We considered genes as potential *mir-1*;Q35 suppressors if less than four worms left the circle upon RNAi knock down of the genes and prioritized candidates when only zero or one worm left. Among the selected candidates, vacuolar ATPase subunits *vha-1*, *vha-8*, *vha-9*, *vha-13*, and *vha-14*, as well as muscle gene *zyx-1*, stood out as molecules suppressing *mir-1* motility (*Figure 2C* and *Figure 2—figure supplement 1D*).

As a parallel approach to identify miR-1 regulatory targets in an unbiased manner, we performed TMT shotgun proteomic analysis, comparing *mir-1* mutants to WT worms on day 1 of adulthood. We identified approximately 1500–2000 proteins in the replicates (*Figure 2D*, *Supplementary file 1f*). Samples showed a high degree of correlation (*Figure 2—figure supplement 1B*) with minimal separation of WT and *mir-1* genotypes, reflecting the subtle changes induced by *mir-1* mutation at the

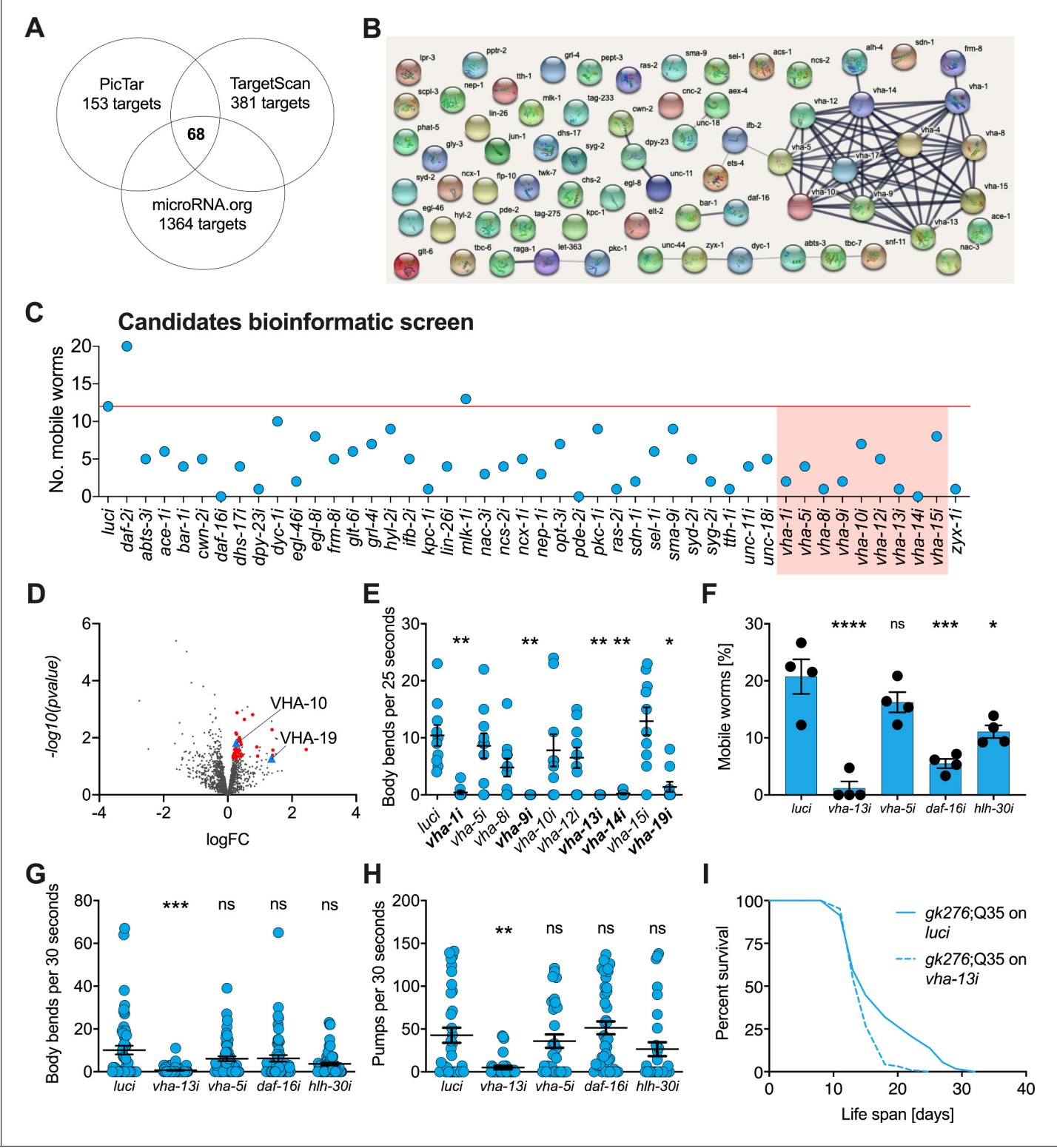

**Figure 2.** v-ATPase subunits are downstream mediators of *mir-1*-induced motility improvement. (**A**) Computational screen for potential miR-1 targets using in silico predictions microRNA.org, TargetScanWorm (TargetScan), and PicTar identifies 68 shared candidates. (**B**) STRING network analysis of predicted miR-1 targets reveals a cluster of v-ATPase subunits. (**C**) Initial RNAi screen of computationally predicted candidates using the circle test on day 8 of adulthood reveals a number of candidates that reduced motility of *mir-1(gk276)*;Q35. 20 worms per RNAi (N = 1). Red line: number of luciferase controls that left the circle. v-ATPases are highlighted in red. (**D**) Volcano plot of proteomic analysis showing differentially regulated proteins in *mir-1(gk276)* vs. N2 animals. VHA-10 and VHA-19 are indicated by blue triangles. Red dots and blue triangles show upregulated proteins tested in

*Figure 2 continued on next page*

Figure 2 continued

circle assay. N = 6 (*mir-1*) and 4 (N2) biological replicates. (E) Effect of v-ATPase subunit RNAi knockdown on *mir-1(gk276)*;Q35 motility as measured in the thrashing assay. Animals are grown on the corresponding RNAi from L4 onwards. N = 1. Mean ± SEM, one-way ANOVA, only significant values are labeled: *p<0.05, **p<0.01. (F, G) Motility assay (circle test) and thrashing assay on day 8 of *mir-1(gk276)*;Q35 worms upon *vha-13, vha-5, daf-16,* and *hlh-30* RNAi knockdown. Control, luciferase RNAi (*luci*). Mean ± SEM of N = 4, one-way ANOVA, *p<0.05, ***p<0.001, ****p<0.0001, ns, not significant. (H) Pumping assay of *mir-1(gk276)*;Q35 worms upon *vha-13, vha-5, daf-16,* and *hlh-30* RNAi knockdown on day 8 of adulthood. Control, luciferase RNAi (*luci*). N = 3, mean ± SEM, one-way ANOVA, **p<0.01, ns, not significant. (I) Life span of *mir-1(gk276)*;Q35 and Q35 worms upon *vha-13* RNAi knockdown. Control, luciferase RNAi (*luci*). One experiment of N = 3, log-rank test: p<0.01.

The online version of this article includes the following figure supplement(s) for figure 2:

**Figure supplement 1.** Bioinformatic and proteomic screens implicate v-ATPase subunits as downstream mediators of *mir-1*-induced motility improvement.

organismal level. For further analysis, we therefore selected the 50 top upregulated proteins with available RNAi clones and tested them for their ability to suppress the motility phenotype using the circle test (*Figure 2—figure supplement 1C, E*, *Supplementary file 1e*), among them *vha-10, vha-19,* and *zyx-1*, which had previously been predicted to harbor *mir-1* binding sites.

Based on the prominence of v-ATPase genes among both the informatic and the proteomic candidates, we systematically tested 10 of them in the thrashing assay, confirming v-ATPase subunits *vha-1, vha-9, vha-13, vha-14,* and *vha-19* as likely mediators of *mir-1*;Q35 motility (*Figure 2E*). We further tested the impact of *vha-5* and *vha-13* on *mir-1*;Q35 phenotypes in more detail, as well as transcription factors *daf-16/FOXO* and *hlh-30/TFEB* as potential miR-1 targets that mediate lysosomal biogenesis. Using the circle test, thrashing assay, and pharyngeal pumping as readout (*Figure 2F–H*), we again verified in particular the role of *vha-13* in reducing *mir-1* motility. In addition, *daf-16/FOXO* and *hlh-30/TFEB* RNAi reduced motility as measured in the circle test, indicating a potential requirement for these transcription factors. We also confirmed that the effect on motility was specific to the *mir-1* background for *vha-5, vha-13,* and *hlh-30* since RNAi of these genes had no significant effect on motility in N2;Q35 worms (*Figure 2—figure supplement 1F*).

In summary, knockdown of 5 out of 10 tested v-ATPase subunits from both the computational and proteomics screen, as well as two transcriptional mediators of lysosome biogenesis, reduced or abolished *mir-1* motility improvement either by circle test or thrashing, revealing these molecules to have congruent physiological effects. Therefore, we focused most of our further efforts on examining v-ATPase regulation by miR-1.

## miR-1 directly regulates VHA-13 protein levels in muscle tissue

MicroRNAs regulate their targets by binding to the 3′UTR of client mRNAs, thereby decreasing RNA stability and translation. If miR-1 regulates v-ATPase subunits, then miR-1 loss would be predicted to de-repress v-ATPase mRNA and protein levels. Because the onset of Q35 aggregation becomes visible by day 4 of adulthood, we examined the mRNA expression of representative v-ATPase subunits at this time. We found that the mRNA levels of *vha-5, vha-10, vha-13,* and *vha-19,* as well as transcription factors *hlh-30*/TFEB and *daf-16*/FOXO, were increased approximately twofold in *mir-1* mutants compared to age-matched WT worms (*Figure 3A*), consistent with the idea that these genes are miR-1 targets.

We further investigated whether miR-1 regulates these gene products at the protein level. We endogenously tagged *vha-5, vha-10, vha-13,* and *vha-19* (N-terminus) with 3xFlag-mNeonGreen-tag as well as *hlh-30* (C-terminus) with mNeonGreen-tag by CRISPR/Cas9 genome editing. Tagging *vha-19* and *vha-10* caused lethality and was not pursued further. Tagged *vha-5* showed significant upregulation in the pharynx in *mir-1(gk276)* mutants compared to WT, but not in vulva muscle (*Figure 3—figure supplement 1A, B*, *Supplementary file 1g*). However, *vha-5* RNAi knockdown had no significant difference in motility compared to controls (*Figure 2E–G*, *Supplementary file 1b*). Further, we saw no obvious regulation of endogenously tagged HLH-30 protein levels in the pharynx in late L4 larvae or day 4 adults (*Figure 4—figure supplement 1A, B*, *Supplementary file 1g*), though we did see increased nuclear localization in the hypodermis in day 4 *mir-1(gk276)* adults (*Figure 4A, B*).

N-terminally tagged *vha-13(syb586)*, however, was viable, showed expression that was regulated, and harbored two potential *mir-1* binding sites in the 3′UTR (*Figure 3F*, *Figure 3—figure supplement 1C*). Moreover, *vha-13* RNAi (*vha-13i*) significantly compromised the motility of *mir-1*;Q35

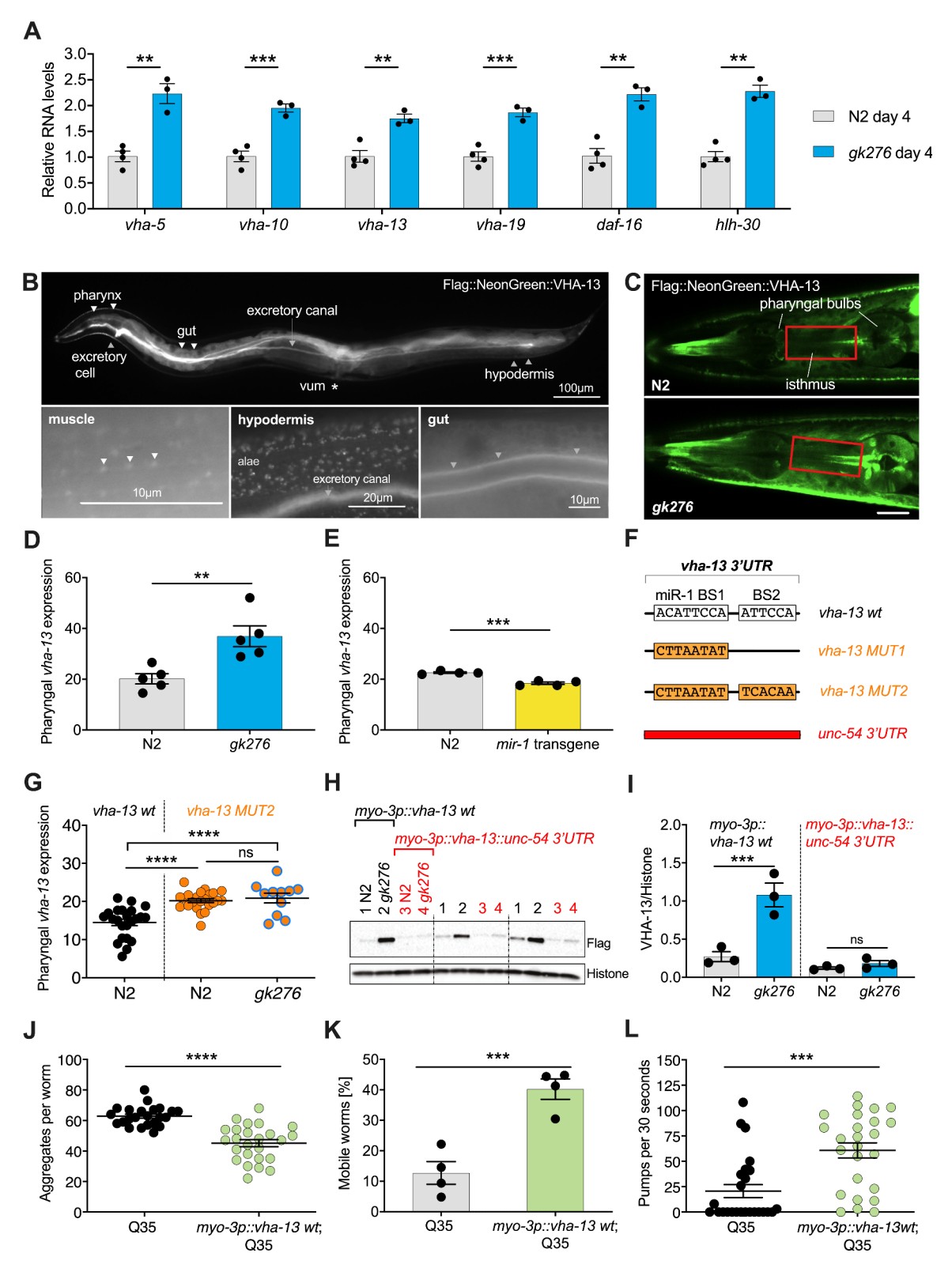

**Figure 3.** miR-1 directly regulates *vha-13* via its 3'UTR in muscle tissue. (**A**) RT-qPCR of *vha-5, vha-10, vha-13,* and *vha-19,* as well as *daf-16* and *hlh-30* mRNA levels in wild-type (WT) (N2) and *mir-1(gk276)* mutants on day 4 of adulthood. Mean ± SEM, N = 3–4, one-way ANOVA, **p<0.01, ***p<0.001. (**B**) Expression pattern of endogenously tagged *3xFlag::mNeonGreen::vha-13* in pharynx, excretory cell and canal, gut, vulva muscle (vum), hypodermis, and muscle (arrowheads indicate dense bodies). (**C**) Confocal images of the head region in worms carrying endogenously tagged *3xFlag::*

*Figure 3 continued on next page*

*Figure 3 continued*

*mNeonGreen::vha-13* in late L4 *mir-1(gk276)* and WT (N2) animals at 25°C. Red rectangle highlights the area of *vha-13* expression in the isthmus used for determination of mNeonGreen intensity. Scale bar 20 μm. (D) Quantification of fluorescent intensity of 3xFlag::mNeonGreen::VHA-13 in the isthmus of indicated genotypes (as shown in C) in late L4 larvae at 25°C. Mean ± SEM of N = 5, t-test, **p<0.01. (E) Quantification of fluorescent intensity of 3xFlag::mNeonGreen::VHA-13 in the isthmus of N2 in the presence or absence of the *mir-1* transgene in late L4 larvae at 25°C. Transgenic worms (*mir-1* transgene) were compared to non-transgenic segregants (N2) of the same strain. Mean ± SEM of N = 4, t-test, ***p<0.001. (F) Graphic showing the 3'UTR of endogenously tagged *3xFlag::mNeonGreen::vha-13. vha-13 wt: vha-13* WT 3'UTR. *vha-13 MUT1*: one miR-1 binding site (BS) is mutated. *vha-13 MUT2*: both miR-1 BSs are mutated. *unc-54 3'UTR*: the *vha-13* 3'UTR is substituted by *unc-54* 3'UTR. Nucleotide sequences of mutated miR-1 BSs are shown below WT. (G) Quantification of fluorescence intensity in the isthmus of L4 larvae with endogenously tagged *3xFlag::mNeonGreen::vha-13* of *vha-13 MUT2* 3'UTR in relation to *vha-13 wt* 3'UTR, in N2 and *mir-1(gk276)* mutant backgrounds at 25°C, using confocal imaging. Mean ± SEM of N = 3, one representative experiment, one-way ANOVA, ****p<0.0001, ns, not significant. (H) Western blot image of late L4 WT and *mir-1* mutants expressing transgenic Flag-tagged *vha-13* in body wall muscle (*myo-3p::vha-13*) with either the *unc-54* 3'UTR (labeled in red), which lacks miR-1 BSs, or the wt *vha-13* 3'UTR, which contains the two miR-1 BSs (labeled in black), immunoblotted with anti-Flag and anti-Histone H3 antibodies. Histone H3 loading control is shown below. Biological replicates (N = 3) separated by dashed lines. (I) Quantification of western blot shown in (H), with flag-tag intensity normalized to histone H3 loading control, mean ± SEM of N = 3, one-way ANOVA, ***p<0.001, ns, not significant. (J) Quantification of aggregates in Q35 worms expressing transgenic *vha-13* in the body wall muscle (*myo-3p::vha-13 wt*;Q35) or non-transgenic segregants (Q35) of the same strain. 25 worms per genotype, mean ± SEM of one representative experiment, N = 4, t-test, ***p<0.001. (K) Motility of Q35 worms expressing transgenic *vha-13* in the body wall muscle (*myo-3p::vha-13 wt*;Q35) or non-transgenic segregants (Q35) of the same strain in circle test. Mean ± SEM. One experiment of N = 2, t-test, ***p<0.001. (L) Pharyngeal pumping rate of *myo-3p::vha-13 wt*;Q35 and Q35 non-transgenic segregants of the same strain. One representative experiment of N = 4, mean ± SEM, t-test, ***p<0.001.

The online version of this article includes the following figure supplement(s) for figure 3:

**Figure supplement 1.** miR-1 regulates *vha-13* via its 3' UTR and its regulation may be evolutionarily conserved.

strains in the RNAi screens (*Figure 2C*), which we further confirmed through thrashing assays and the circle test (*Figure 2E–G*). *vha-13i* also significantly reduced pharyngeal pumping and life span in the *mir-1*;Q35 background compared to *luci* RNAi controls (*Figure 2H, I*, *Supplementary file 1a, b*), yet had relatively minor effects on the motility of N2;Q35 itself (*Figure 2—figure supplement 1F*). We therefore focused on *vha-13* as a promising candidate to pursue in depth for regulatory interactions with miR-1.

We first characterized the expression pattern of *vha-13(syb586[3xFlag::mNeonGreen::vha-13])* in more detail. The *3xFlag::mNeonGreen::vha-13* was strongly expressed in the excretory cell, canal, hypodermis, as well as gut (*Figure 3B*). In the hypodermis, 3xFlag::mNeonGreen::VHA-13 was localized in discrete foci (*Figure 3B*, *Figure 3—figure supplement 1D*). Notably, 3xFlag::mNeonGreen::VHA-13 also resided in miR-1-expressing tissues of body wall muscle, sex muscles, and pharynx, where it was more weakly expressed. Within muscle cells, VHA-13 localized to dense bodies (analogous to vertebrate Z disks) and intervening cytosol (*Figure 3B*, *Figure 3—figure supplement 1D*).

Because *mir-1* is expressed in muscle tissues, we focused on quantifying the expression of *vha-13* in these tissues. Strikingly, we observed that *vha-13* levels were upregulated in the pharyngeal isthmus of *mir-1(gk276)* mutants (*Figure 3C, D*, *Supplementary file 1g*), while conversely, overexpression of the *mir-1* transgene decreased levels in this tissue relative to WT (*Figure 3E*, *Supplementary file 1g*). To further investigate whether miR-1 directly regulates *vha-13*, we altered two predicted miR-1 binding sites (BS) in the 3'UTR (*Figure 3F*). We first mutated miR-1 binding site 1 (*vha-13* BS1 [*syb504*], at position 188–195 of *vha-13* 3'UTR), and then additionally miR-1 binding site 2 (*vha-13* BS2 [*syb2180*], at position 253–258 of *vha-13* 3'UTR) in the endogenously tagged *vha-13(syb586)* strain. When only BS1 was mutated (MUT1), we still saw residual regulation of *vha-13* by miR-1 (*Figure 3—figure supplement 1E*, *Supplementary file 1g*). Mutating both miR-1 binding sites (BS1,2: MUT2), however, de-repressed VHA-13 levels in the isthmus in the WT background (*Figure 3G*). In double mutants containing both *mir-1(gk276)* and *vha-13* BS1,2 mutations, no clear regulation of *vha-13* expression was observed compared to the BS1,2 mutations alone (*Figure 3G*), suggesting that miR-1 modulates pharyngeal *vha-13* expression through both potential miR-1 binding sites.

We next sought to characterize *vha-13* regulation in the body wall muscle. However, it was not possible to accurately measure expression in this tissue using the *3xFlag::mNeonGreen::vha-13* strain due to the high expression levels in the adjacent hypodermis and gut. We therefore generated transgenic worms expressing Flag-tagged *vha-13* containing its endogenous 3'UTR (*3xFlag::vha-13::vha-13 3'UTR*) under control of the muscle-specific *myo-3* promoter. Consistent with results seen in

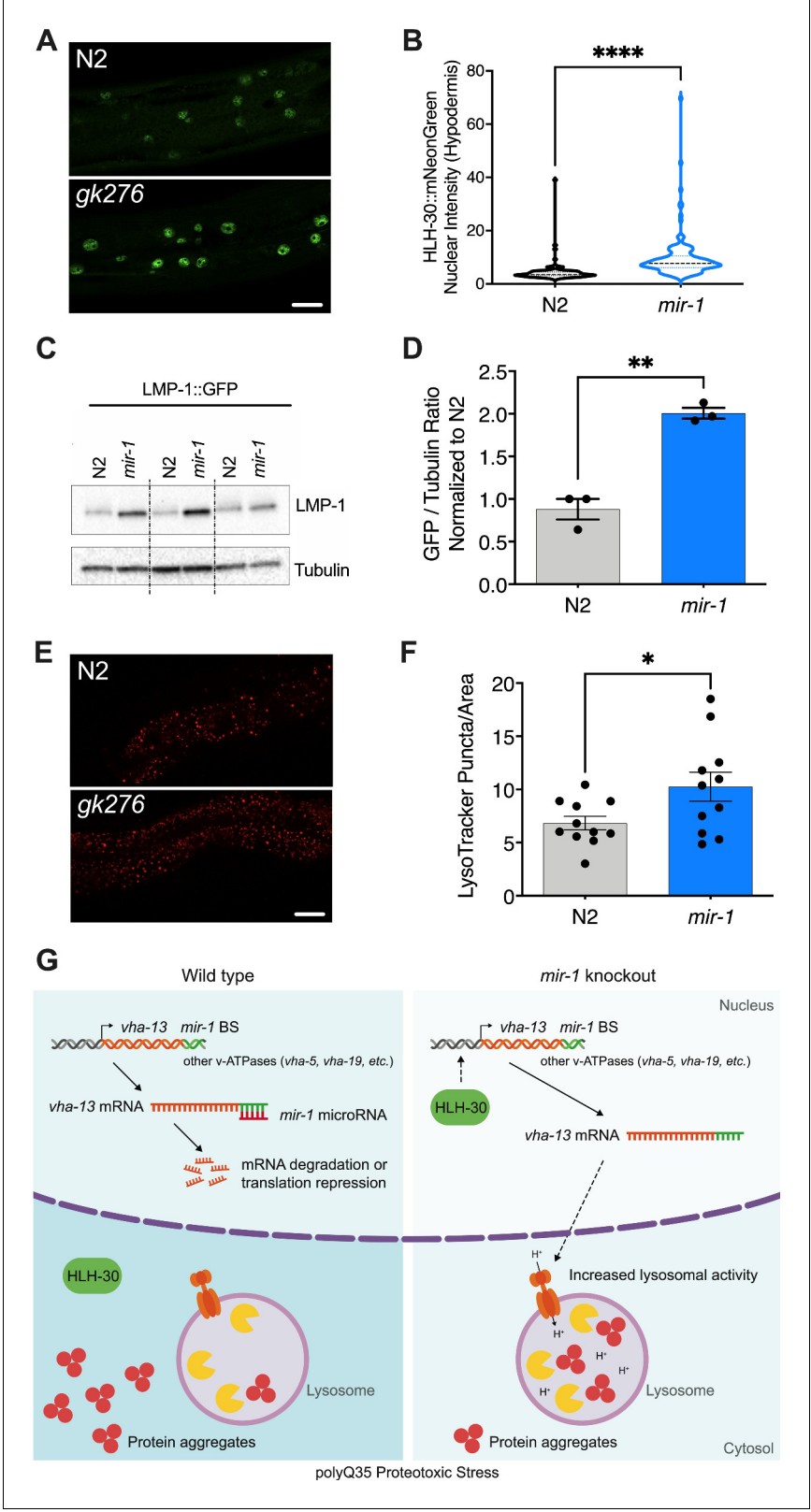

**Figure 4.** *mir-1* mutation enhances lysosomal biogenesis and acidification. (**A**) Fluorescent image comparing HLH-30::mNeonGreen nuclear localization in the hypodermis of the *mir-1(gk276)* and wild-type (WT) (N2) backgrounds at day 4 of adulthood, maintained at 25°C. Scale bar 20 μm. (**B**) Quantitation of nuclear localization in (**A**). Violin plot, mean ± SEM of one representative experiment of N = 3, t-test, **** p<0.0001. (**C**) Western blot of LMP-1::
*Figure 4 continued on next page*

*Figure 4 continued*

GFP in WT and *mir-1* mutants at the L4 stage, immunoblotted with anti-GFP or anti-α-tubulin antibodies. Biological replicates (N = 3) separated by dashed lines. (D) Quantification of the western blot in (C), normalized to α-tubulin loading control. N = 3 BR, line and error bars indicate mean ± SEM, t-test, **p<0.01. (E) Representative images of lysotracker staining in WT and *mir-1(gk276)* mutants at day 4 of adulthood at 25°C. Scale bar 20 µm. (F) Quantification of lysotracker images using a predefined squared area approximately spanning the second to fourth gut cell. Quantification was performed using ImageJ. N = 3 BR, line and error bars indicate mean ± SEM of combined experiments, t-test, *p<0.05. (G) Working model. miR-1 normally limits proteoprotective pathways, downregulating the expression of *vha-13,* other v-ATPases and factors by binding miR-1 binding site(s) in the 3′UTR (miR-1 BS) of the corresponding mRNA (WT, polyQ35 proteotoxic stress). Loss of miR-1 (*mir-1* knockout, polyQ35 proteotoxic stress) results in de-repression of *vha-13* and elevated lysosomal activity. Nuclear localization of HLH-30/TFEB, a master regulator of lysosome biogenesis, is also enhanced, collectively resulting in reduced number of aggregates under polyQ35 proteotoxic stress conditions.

The online version of this article includes the following figure supplement(s) for figure 4:

**Figure supplement 1.** *mir-1* mutation does not affect total expression of HLH-30 in the pharynx.

pharyngeal muscle, *vha-13* protein levels in the body wall muscle were also upregulated in *mir-1* mutants compared to WT, as measured by western blots (*Figure 3H, I*). Exchanging the *vha-13* 3′UTR with the *unc-54* 3′UTR lacking miR-1 binding sites (*vha-13::unc-54 3′UTR*) blunted regulation by miR-1 (*Figure 3F, I*), indicating that *vha-13* is repressed by miR-1 via its 3′UTR in both body wall and pharyngeal muscle.

Since *mir-1* mutation resulted in *vha-13* upregulation, we asked whether *vha-13* overexpression was sufficient to yield phenotypes similar to *mir-1* mutants. In accord with this idea, overexpressing *vha-13* in the muscle of Q35 worms significantly reduced aggregate number, improved motility, and enhanced pharyngeal pumping ability (*Figure 3J–L*, *Figure 3—figure supplement 1F*, *Supplementary file 1b*). It also significantly increased the life span of Q35 animals compared to non-transgenic controls in two out of five experiments (*Figure 3—figure supplement 1G*, *Supplementary file 1a*). Altogether these data suggest that miR-1 and VHA-13 work in the same regulatory pathway to influence muscle physiology.

Since the regulation of *vha-13* by miR-1 was robust and functionally relevant in *C. elegans*, we wondered whether this regulation may be evolutionarily conserved. We analyzed v-ATPase expression in transcriptome data from previously published muscle-specific miR-1/miR-133a double knockout mice (*Wüst et al., 2018*) and miR-1-1/miR-1-2 double knockout mice (*Wei et al., 2014*). In muscle-specific miR-1/133a double knockout mice, *Atp6v1a/vha-13* and *Atp6v1h/vha-15* are significantly upregulated by 2.6-fold and 2.1-fold, respectively (*Figure 3—figure supplement 1H*). In miR-1-1/miR-1-2 double knockout mice, *Atp6v1a/vha-13* is significantly upregulated by 1.49-fold (*Figure 3—figure supplement 1I*). These findings suggest that miR-1-dependent regulation of *vha-13/Atp6v1a* may be conserved during evolution.

## miR-1 affects lysosomal biogenesis

As the v-ATPase is an integral component of the lysosome, we asked whether miR-1 generally affects lysosomal structure and function. To test this idea, we first used a *lmp-1p:lmp-1::gfp* reporter strain. LMP-1 protein localizes to the membrane of lysosomes and is a marker for lysosomal biogenesis (*Hermann et al., 2005*). LMP-1 protein levels were significantly increased in *mir-1* mutants compared to WT control, as measured by western blot analysis (*Figure 4C, D*). The v-ATPase hydrolyzes ATP to pump protons across the membrane, resulting in acidification of the lysosome lumen (*Beyenbach and Wieczorek, 2006*). We therefore asked whether *mir-1* mutants affect the number of acidified lysosomes. Using Lysotracker Red, a dye that targets acidic membranous structures such as lysosomes (*Chazotte, 2011*), we observed an increase in the number of acidified puncta in *mir-1* mutants at day 4 of adulthood (*Figure 4E, F*). Due to technical limitations, lysotracker staining could only be observed in the worm intestine. This result leaves open the question as to whether lysosomes are also regulated in the muscle and whether *mir-1*, being expressed in muscle tissue, might have a cell non-autonomous effect. Nonetheless, the overall increase in lysosome biogenesis is consistent with the upregulation of lysosomal components, such as the v-ATPase subunits, and a possible cause of miR-1-dependent regulation of proteostasis.

## Discussion

Striated muscle is one of the most highly ordered tissues in the body, with molecular components organized in a lattice of contractile elements and attachments. This molecular apparatus is exposed to high energy and force during contraction, inflicting molecular damage requiring constant repair. Further, muscle is subject to growth, metabolic, and stress signaling pathways as well as neural inputs that also promote remodeling. Underlying muscle plasticity is the fine-tuned control of proteostasis, including protein synthesis, folding, trafficking, and turnover, which must be precisely orchestrated to maintain muscle structure and function (*Demontis et al., 2013*). A decline in these processes during aging leads to diminished muscle performance and frailty, yet it remains elusive how various aspects of muscle proteostasis are coordinated.

In this work, we discovered that the muscle-enriched microRNA miR-1 plays an important role in regulating muscle homeostasis via vacuolar ATPase function. Loss of *mir-1* ameliorates age-related decline in motility induced by models of aggregate-prone polyQ35. By inference, *mir-1* normally limits the proteoprotective effect in the polyQ35 model. Computational and proteomic screens identified v-ATPase subunits as highly enriched targets of miR-1 regulation, suggestive of coordinate regulation, and whose downregulation reduced polyQ35 proteoprotection. Expression studies confirmed that several subunits (e.g., *vha-5, vha-10, vha-13, vha-19*) showed miR-1-dependent regulation. In particular, we demonstrated that *vha-13* expression is downregulated by miR-1 in muscle tissues via two distinct binding sites in its 3′UTR. Moreover, VHA-13 links miR-1 with muscle homeostasis since *vha-13* downregulation abolished the improved mid-life motility, pharyngeal pumping, and life span of *mir-1*;Q35 strains, while *vha-13* overexpression was sufficient to enhance these properties, similar to *mir-1* mutation. In accord with our findings, immunoprecipitation and sequencing of microRNA complexes revealed a number of v-ATPase subunit mRNAs, including *vha-13, vha-4, vha-10,* and *vha-17* as physically associated with *C. elegans* miR-1 (*Grosswendt et al., 2014*). Interestingly, mammalian homolog of VHA-13, ATP6V1A, as well as several other v-ATPase subunits, harbor predicted miR-1 binding sites (*Stark et al., 2005*). Transcriptome data from mice indicate that *Atp6v1a* is upregulated upon miR-1-1/miR-1–2 deletion or miR-1/miR-133a deletion (*Wei et al., 2014*; *Wüst et al., 2018*). These findings complement our results in *C. elegans* and suggest that the miR-1-ATP6V1A regulatory module could be conserved in evolution.

The v-ATPase is a multisubunit enzyme that acidifies the endolysosomal lumen to control a plethora of cellular activities. Acidification regulates protein trafficking, endocytic recycling, synaptic vesicle loading, and autophagy, as well as the activity of multiple acid hydrolases and nutrient and ion transporters. Moreover, the v-ATPase itself serves as a docking site for regulating mTOR and AMPK complexes and affects metabolism (*Settembre et al., 2013*; *Zhang et al., 2014*). Our studies highlight the importance of v-ATPase activity to muscle performance. A handful of studies ascribe a role for the v-ATPase in muscle. In mammalian cardiomyocytes, lipid loading inhibits the v-ATPase, leading to a decline in contractile function that could contribute to muscle deficits in diabetes (*Wang et al., 2020*). Lesions in VMA21 that disrupt v-ATPase assembly have also been shown to cause myopathies (*Dowling et al., 2015*). Conceivably the v-ATPase could play an important role in protein turnover and remodeling of muscle structure but could also impact muscle homeostasis through metabolic wiring or protein trafficking.

Surprisingly we also found that some *mir-1* phenotypes were not strictly limited to muscle since we observed a global increase in levels of lysosomal LMP-1::GFP in western blots, increased lysosomal acidification in the intestine, as well as enhanced nuclear localization of HLH-30/TFEB in the hypodermis. Given that *mir-1* is muscle-expressed, this observation could suggest cell non-autonomous regulation of lysosomal biogenesis and associated activities by miR-1, acting either directly or indirectly. Conceivably, miR-1 is secreted from muscle to affect physiology in other tissues. In mammals, miR-1 has been identified as a circulating microRNA found in serum exosomes upon exercise stress or cardiac infarction, suggesting that it could act systemically (*Cheng et al., 2019*; *D'Souza et al., 2018*). In this light, it is intriguing that the v-ATPase itself is implicated in exosomal activity (*Liégeois et al., 2006*). Alternately, miR-1 could act indirectly through production of a myokine that affects distal tissues.

miR-1 has been implicated in regulating a number of targets and physiologic processes. In mammals, miR-1 and its homolog miR-133 are essential to cardiac development and function, where they have been shown to regulate Notch ligand Dll1, the GTPase dynamin 2 (DNM2), Frizzled-7 (FZD7),

and fibroblast growth factor receptor substrate 2 (FRS2) as targets (*Ivey et al., 2008*; *Liu et al., 2011*; *Mitchelson and Qin, 2015*). Surprisingly knockout of mammalian *mir1/mir133* specifically in muscle has little overt effect on muscle structure, but rather regulates the developmental transition from glycolytic to oxidative metabolism via the Mef2/Dlk1-Dio3 axis, affecting running endurance (*Wüst et al., 2018*). Similarly, *C. elegans* miR-1 regulates *mef-2*, in this case affecting retrograde signaling at the neuromuscular junction (*Simon et al., 2008*). We also observed that animals lacking *mir-1* have little overt change in muscle structure or function alone, though we saw upregulation of a number of muscle proteins and downregulation of mitochondrial proteins in our proteomic analysis. Indeed, most microRNA knockouts do not lead to observable phenotypes in *C. elegans* (*Miska et al., 2007*), suggesting that microRNAs often work redundantly or in response to stress to fine-tune gene expression.

We identified *mir-1* as a suppressor of the muscle-specific polyQ35 proteotoxicity model, reducing protein aggregation and having beneficial effects on late-life motility, pumping rates (day 8 of adulthood), and life span. Recently, Pocock and colleagues reported that miR-1 downregulates the Rab GTPase TBC-7/TBC1D15, thereby promoting autophagic flux in worms and human cells (*Nehammer et al., 2019*). In *C. elegans,* they observed that *mir-1(+)* ameliorates polyQ40 proteotoxicity, seemingly contradicting our results. We found *mir-1* deletion to reduce aggregate numbers in polyQ35, polyQ40, and α-syn models, while effects on motility varied among the three models. The relationship between aggregation and cellular toxicity remains controversial, and the uncoupling of aggregation and toxicity is observed in various disease models. Mammalian studies suggest context-dependent *mir-1* effects, where *mir-1* either promotes or inhibits autophagy (*Ejlerskov et al., 2020*; *Hua et al., 2018*; *Xu et al., 2020*). Whether miR-1 is proteoprotective or limits proteoprotection in *C. elegans* could hinge on many factors. Because microRNAs generally fine-tune regulation and often work in feedback circuits, their impact on physiology can be subtle, and small differences in culture conditions, genetic background, metabolic state, and aggregation onset during development or aging could give rise to divergent outcomes.

Among the putative miR-1 targets that we identified in *C. elegans* are the pro-longevity transcription factors DAF-16/FOXO and HLH-30/TFEB, which regulate lysosome biogenesis, proteostasis, and metabolism, and act in insulin/IGF and mTOR signaling pathways (*Lin et al., 2018*; *O'Rourke and Ruvkun, 2013*; *Settembre et al., 2011*). *mir-1* mutation led to an upregulation of their mRNA during adulthood and enhanced nuclear localization of HLH-30. Further RNAi knockdown of these factors modestly diminished the motility improvement of *mir-1* mutants in the circle test, consistent with roles in a miR-1 signaling pathway. Whether these transcription factors are direct targets of miR-1 regulation remains to be seen. Notably, miR-1-predicted targets also include pro-aging components of mTOR signaling such as *let-363/*mTOR and *raga-1/*RagA (targetscan.org), which also regulate the activity or localization of these transcription factors. Hence, the balance or timing of pro- and anti-aging factors could also differentially influence the physiological phenotype at a systemic level. Interestingly, VHA-13/ATP6V1A subunit has been shown to regulate mTOR lysosomal recruitment and activity (*Chung et al., 2019*) and mTOR signaling regulates the activity of TFEB at the lysosomal surface (*Settembre et al., 2013*). Upregulation of v-ATPase activity is also associated with the clearance of oxidized protein and rejuvenation of the *C. elegans* germline (*Bohnert and Kenyon, 2017*). Thus, in the future it will be interesting to further unravel the miR-1 molecular circuitry surrounding lysosomal function and proteostasis, and see whether miR-1 similarly regulates v-ATPase function in humans.

## Materials and methods

### Key resources table

| Reagent type (species) or resource | Designation | Source or reference | Identifiers | Additional information |
|---|---|---|---|---|
| Gene (*Caenorhabditis elegans*) | *mir-1* | WormBase | WBGene00003260 | |

*Continued on next page*

*Continued*

| Reagent type (species) or resource | Designation | Source or reference | Identifiers | Additional information |
|---|---|---|---|---|
| Gene (*Caenorhabditis elegans*) | *vha-13* | WormBase | WBGene00013025 | |
| Gene (*Caenorhabditis elegans*) | *vha-5* | WormBase | WBGene00006914 | |
| Gene (*Caenorhabditis elegans*) | *hlh-30* | WormBase | WBGene00020930 | |
| Gene (*Caenorhabditis elegans*) | *daf-16* | WormBase | WBGene00000912 | |
| Strain, strain background (*Caenorhabditis elegans*) | N2 Bristol | CGC | N2 Bristol RRID:WB-STRAIN: WBStrain00000001 | N2 WT |
| Strain, strain background (*Caenorhabditis elegans*) | *mir-1(gk276)* I | This paper | AA2508 RRID:WB-STRAIN: WBStrain00035886 | |
| Strain, strain background (*Caenorhabditis elegans*) | *mir-1(dh1111)* I | This paper | AA4575 | |
| Strain, strain background (*Caenorhabditis elegans*) | *rmIs132[unc-54p::Q35::YFP]* I | CGC | AM140 RRID:WB-STRAIN: WBStrain00000182 | N2;Q35 |
| Strain, strain background (*Caenorhabditis elegans*) | *mir-1(gk276); rmIs132[unc-54p::Q35::YFP]* I | This paper | AA4403 | *gk276*;Q35 |
| Strain, strain background (*Caenorhabditis elegans*) | *mir-1(dh1111) r mIs132[unc-54p::Q35::YFP]* I | This paper | AA4577 | *dh1111*;Q35 |
| Strain, strain background (*Caenorhabditis elegans*) | N2; *dhEx965[mir-1p::mir-1, myo-2p::mCherry]* | This paper | AA3275 | *mir-1* transgene |
| Strain, strain background (*Caenorhabditis elegans*) | *mir-1(gk276)* I; *rmIs132[unc-54p::Q35::YFP]; dhEx965[mir-1p::mir-1, myo-2p::mCherry]* | This paper | AA4810 | |
| Strain, strain background (*Caenorhabditis elegans*) | N2; *dhEx1206[myo 3p::flag::HA::mCherry::vha-13cDNA::unc-54 3′UTR, myo-2p::GFP]* | This paper | AA4865 | *myo3p::vha-13::unc-54 3′UTR* |
| Strain, strain background (*Caenorhabditis elegans*) | *mir-1(gk276)* I; *dhEx1206[myo-3p::flag::HA::mCherry::vha-13cDNA::unc-54 3′UTR, myo-2p::GFP]* | This paper | AA4866 | |
| Strain, strain background (*Caenorhabditis elegans*) | N2; *dhEx1207[myo-3p::flag::HA::mCherry::vha-13cDNA::vha-13 3′UTR, myo-2p::GFP]* | This paper | AA5067 | *myo-3p::vha-13 wt 3′UTR* |

*Continued on next page*

*Continued*

| Reagent type (species) or resource | Designation | Source or reference | Identifiers | Additional information |
|---|---|---|---|---|
| Strain, strain background (*Caenorhabditis elegans*) | *mir-1(gk276)* I; *dhEx1207[myo-3p::flag::HA::mCherry::vha-13cDNA::vha-13 3'UTR, myo-2p::GFP]* | This paper | AA5068 | |
| Strain, strain background (*Caenorhabditis elegans*) | *vha-13(syb586 [3xFlag::mNeon Green::vha-13])* V | SunyBiotech | PHX586 | *vha-13 wt 3'UTR* |
| Strain, strain background (*Caenorhabditis elegans*) | *mir-1(gk276)* I; *vha-13(syb586 [3xFlag::mNeon Green::vha-13])* V | This paper | AA4813 | |
| Strain, strain background (*Caenorhabditis elegans*) | *vha-13(syb586 [3xFlag::mNeon Green::vha-13])* V; *dhEx965[mir-1p::mir-1, myo-2p::mCherry]* | This paper | AA4850 | *vha-13 wt 3'UTR + mir-1* transgene |
| Strain, strain background (*Caenorhabditis elegans*) | *vha-13(syb587, syb504[3xFlag::mNeon Green::vha-13 miR-1 BS1 mutated])* V | SunyBiotech | PHX587 | *vha-13 MUT1 3'UTR* One *miR-1* BS mutated |
| Strain, strain background (*Caenorhabditis elegans*) | *mir-1(gk276)* I; *vha-13(syb587, syb504[3xFlag::mNeon Green::vha-13 miR-1 BS1 mutated])* V | This paper | AA4184 | |
| Strain, strain background (*Caenorhabditis elegans*) | *vha-13(syb2180, syb587,syb504 [3xFlag::mNeon Green::vha-13 miR-1 BS1,2 mutated])* V | SunyBiotech | PHX2180 | *vha-13 MUT2 3'UTR* Two *miR-1* BSs mutated |
| Strain, strain background (*Caenorhabditis elegans*) | *mir-1(gk276)* I; *vha-13(syb2180,syb587, syb504[3xFlag::mNeon Green::vha-13 miR-1 BS,2 mutated])* V | This paper | AA5123 | |
| Strain, strain background (*Caenorhabditis elegans*) | *vha-5(syb1093 [3xFlag::mNeon Green::vha-5])* IV | SunyBiotech | PHX1093 | |
| Strain, strain background (*Caenorhabditis elegans*) | *mir-1(gk276)* I; *vha-5(syb1093 [3xFlag::mNeon Green::vha-5])* IV | This paper | AA5069 | |
| Strain, strain background (*Caenorhabditis elegans*) | *hlh-30(syb809[hlh-30::mNeonGreen])* IV | SunyBiotech | PHX809 | |
| Strain, strain background (*Caenorhabditis elegans*) | *mir-1(gk276)* I; *hlh-30(syb809[hlh-30::mNeonGreen])* IV | This paper | AA5195 | |
| Strain, strain background (*Caenorhabditis elegans*) | *rmIs133[P(unc-54) Q40::YFP]* | CGC | AM141 RRID:WB-STRAIN: WBStrain00000183 | N2;Q40 |

*Continued on next page*

*Continued*

| Reagent type (species) or resource | Designation | Source or reference | Identifiers | Additional information |
|---|---|---|---|---|
| Strain, strain background (*Caenorhabditis elegans*) | *mir-1(gk276)* I; *rmIs133[P(unc-54) Q40::YFP]* | This paper | AA3112 | *gk276;Q40* |
| Strain, strain background (*Caenorhabditis elegans*) | N2; *pkIs2386[Punc-54::alphasynuclein::YFP + unc-119(+)]* | This paper | AA1311 | |
| Strain, strain background (*Caenorhabditis elegans*) | *mir-1(gk276)* I; *pkIs2386[Punc-54::alphasynuclein::YFP + unc-119(+)]* | This paper | AA5270 | |
| Strain, strain background (*Escherichia coli*) | OP50 | CGC | | |
| Strain, strain background (*Escherichia coli*) | DH5α | Life Technologies | | |
| Other | RNAi | Source BioScience, *C. elegans* RNAi Collection – Ahringer | Multiple | *Supplementary file 1e* |
| Other | RNAi | Source BioScience, *C. elegans* ORF-RNAi Resource – Vidal | Multiple | *Supplementary file 1e* |
| Other | *hlh-30* RNAi | Source BioScience, *C. elegans* RNAi Collection – Ahringer | W02C12.3 | *hlh-30i* |
| Other | *daf-16* RNAi | Source BioScience, *C. elegans* RNAi Collection – Ahringer | R13H8.1 | *daf-16i* |
| Other | *vha-13* RNAi | Source BioScience, *C. elegans* RNAi Collection – Ahringer | Y49A3A.2 | *vha-13i* |
| Other | *vha-5* RNAi | Source BioScience, *C. elegans* RNAi Collection – Ahringer | F35H10.4 | *vha-5i* |
| Recombinant DNA reagent | Plasmid L3781 | FireLab | | Worm Expression vector |
| Recombinant DNA reagent | Plasmid pDEST R4-R3 | Invitrogen | | MultiSite Gateway destination vector |
| Recombinant DNA reagent | Plasmid *myo3p::flag::HA::mCherry::vha-13cDNA::unc-54 3'UTR* | This paper | BA4331 | Vector pDEST R4-R3 |
| Recombinant DNA reagent | Plasmid *myo3p::flag::HA::mCherry::vha-13cDNA::vha-13 3'UTR* | This paper | BA4334 | Vector pDEST R4-R3 |
| Recombinant DNA reagent | Plasmid pPD122.11 | FireLab | | Worm Expression vector |
| Sequence-based reagent | Primer | This paper | | *Supplementary file 1h* |
| Commercial assay, kit | TaqMan assay for mir-1 | Life Technologies | Assay ID 000385 | |

*Continued on next page*

*Continued*

| Reagent type (species) or resource | Designation | Source or reference | Identifiers | Additional information |
|---|---|---|---|---|
| Antibody | Anti-GFP (mouse monoclonal) | Living Colors | JL-8 RRID:AB_10013427 | 1:2000 |
| Antibody | Anti-Histone H3 (rabbit polyclonal) | Abcam | ab1791 RRID:AB_302613 | 1:5000 |
| Antibody | Anti-α-Tubulin (mouse monoclonal) | Sigma | T6199 RRID:AB_477583 | 1:5000 |
| Antibody | Anti-FLAG (mouse monoclonal) | Sigma | F1804 RRID:AB_262044 | 1:2000 |
| Antibody | Anti-mouse IgG (goat polyclonal coupled with horseradish peroxidase) | Invitrogen | G-21040 RRID:AB_2536527 | 1:5000 |
| Antibody | Anti-rabbit IgG (goat polyclonal coupled with horseradish peroxidase) | Invitrogen | G-21234 RRID:AB_2536530 | 1:5000 |
| Chemical compound, drug | LysoTracker Red | Life Technologies | LysoTracker Red DND-99 | |

## *C. elegans* strains and culture

*C. elegans* strains were maintained at 20°C on nematode growth medium (NGM) plates seeded with a lawn of *Escherichia coli* strain OP50, unless noted otherwise. Strains are listed in the Key resources table.

## Molecular cloning

All restriction digest reactions were performed with enzymes provided by NEB according to the user's manual. T4 DNA Ligase (NEB) was used for ligation reactions. Chemically competent DH5α *E. coli* (Life Technologies) was used for transformation following the manufacturer's instructions. QIAprep Miniprep or Midiprep Kits (Qiagen) were used for plasmid purification. Cloning was verified by PCR followed by gel electrophoresis and sequencing.

To make the rescuing *mir-1* transgene, primer pair 'celmir1fwd2/rvs2' was used to insert the *mir-1* coding region into vector L3781 downstream of *gfp*. The *mir-1* promoter was then cloned 5' to *gfp* using the primer pair 'm1p fwd/rvs'. To make muscle-expressed *vha-13* constructs, the *vha-13* cDNA was amplified with Kpn1 overhangs using primers vha-13 fwd/rvs and cloned into vector pDEST R4-R3 to give *myo-3p::flag::HA::mCherry::vha-13cDNA::unc-54* 3'UTR. The *unc-54* 3'UTR was excised with Not1/BglI and replaced with the *vha-13* 3'UTR using primers vha-13U fwd/rvs to generate *myo-3p::flag::HA::mCherry::vha-13cDNA::vha-13* 3'UTR. Primers and plasmids are listed in the Key resources table and *Supplementary file 1h*.

## Generation of transgenic worm strains

Transgenic worms containing extrachromosomal arrays were generated by microinjection. To generate the *myo-3p::flag::HA::mCherry::vha-13cDNA::unc-54* 3'UTR strain, a mix of *myo-3p::flag::HA::mCherry::vha-13cDNA::unc-54* 3'UTR DNA (40 ng/ µl), *myo-2p::gfp* co-transformation marker (5 ng/ µl plasmid pPD122.11), and fill DNA (TOPO empty vector, 55 ng/µl) was injected into young N2 adults using an Axio Imager Z1 microscope (Zeiss) with a manual micromanipulator (Narishige) connected to a microinjector (FemtoJet 4x, Eppendorf). We obtained the strain AA4865: N2; *dhEx1206 [myo-3p::flag::HA::mCherry::vha-13cDNA::unc-54* 3'UTR, *myo-2p::gfp*], which was then used to cross the transgene into other genetic backgrounds. A similar strategy was used to create AA5067, N2; dh*Ex1207[myo-3p::flag::HA::mCherry::vha-13cDNA::vha-13* 3'UTR, *myo-2p::GFP*].

The *mir-1* transgene strain was generated by injection mix containing *mir-1p::mir-1* plasmid, *myo-2p::mCherry* co-transformation marker (5 ng/µl plasmid pPD122.11) and fill DNA (L3781 empty vector). The resultant strain AA3275 (N2; *dhEx965[mir-1p::mir-1, myo-2p::mCherry]*) was then used to

cross the *dhEx965* transgene to *mir-1(gk276); rmIs132[p(unc-54) Q35::YFP]* to give *mir-1(gk276); rmIs132[p(unc-54) Q35::YFP]; dhEx965[mir-1p::mir-1, myo-2p::mCherry]*, AA4810. For transgenic worm strains, non-transgenic worms of the same strain were used as controls.

The *mir-1* deletion allele *dh1111* was generated using CRISPR/Cas9 mutagenesis. We designed CRISPR guides using the EnGen sgRNA Designer (https://sgrna.neb.com/#!/sgrna), synthesized guides with the Engen sgRNA synthesis kit and analyzed them by gel electrophoresis and tape station. We injected worms with an injection mix containing Cas9 EnGen (NEB), four sgRNAs against *mir-1* (AAGAAGTATGTAGAACGGGG, GTAAAGAAGTATGTAGAACG, TATAGAGTAGAATTGAATCT, ATATAGAGTAGAATTGAATC), one sgRNA against *dpy-10* (CGCTACCATAGGCACCACG), KCl, HEPES pH 7.4, and water. Prior to injection we incubated the mixture for 10 min at 37°C to allow activation of Cas9. Following injection, we singled out worms with Dpy and Rol phenotypes in the F1 generation and genotyped them for *mir-1* deletion using *mir-1* genotyping primers in the F2 (*Supplementary file 1h*). We sequenced the PCR products of candidate worms with Sanger sequencing and verified that the deletions resulted in loss of *mir-1* expression using TaqMan-based quantification of mature miRNA levels.

Endogenous fluorescently tagged strains were generated by tagging *vha-5* or *vha-13* with 3xFlag-mNeonGreen tag at the N-terminus using CRISPR–Cas9 (SunyBiotech). The *vha-13* 3'UTR mutants BS1 and BS2 were generated using CRISPR–Cas9 by further mutating one or both putative miR-1 binding sites in the 3'UTR of the endogenous FLAG-mNeonGreen-tagged *vha-13 gene* (SunyBiotech). PHX809 *hlh-30::mNeonGreen* endogenously tagged *hlh-30* was generated by placing *mNeon-Green* at the C-terminus using CRISPR–Cas9 (SunyBiotech).

## Determination of progeny number

Single worms were maintained on an NGM agar plate and transferred every day until the reproductive period was complete. The number of F1 progeny per individual worm was counted at the L4 or young adult stage. Experiments were repeated at least three times.

## Quantitation of polyQ and α-synuclein aggregates

Whole-worm images of day 4 adults (Q34), L4s (Q40), or day 1 adults (α-synuclein) were taken with an Axiocam 506 mono (Zeiss) camera using the 5×, 10×, or 20× objective of a Zeiss Axio Imager Z1 microscope at defined exposure time (Q35: 20 ms, Q40: 10 ms, α-synuclein: 10 ms). Aggregate numbers were evaluated from the photos. Due to the smaller size and relatively lower number of aggregates in the α-synuclein model, a maximum intensity projection was created from a Z-stack of the head region and aggregates present from the nose tip to the posterior bulb were counted. Genotypes of the samples were blinded during the counting. Aggregates were defined as discrete structures or puncta above background.

## Motility, thrashing, and pumping behavior

To analyze worm motility by the circle test, 20–25 worms of defined age (day 3 to day 14 of adulthood) were placed in the center of a 6 cm agar plate with bacteria, marked with 1 cm circle on the bottom. After a defined time period (1 or 30 min), the number of worms that left the circle was counted. To determine thrashing rate, individual worms were transferred to M9 buffer and the number of body bends in a 20–60 s interval was scored. Pharyngeal pumping rates were measured by counting grinder contraction in the terminal bulb in 20–30 s intervals. Genotypes were blinded during the experiments.

## CeleST assay

The *C. elegans* Swim Test (CeleST) assay was used to assess animal motility while swimming; assays were conducted as described previously (*Ibáñez-Ventoso et al., 2016*; *Restif et al., 2014*). Four or five animals were picked and placed into the swimming arena, which was a 50 µl aliquot of M9 buffer inside a 10 mm pre-printed ring on the surface of a glass microscope slide (Thermo Fisher Scientific). Images of the animals within the swim arena were acquired with a LEICA MDG41 microscope at 1× magnification with a Leica DFC3000G camera. Image sequences of 30 s in duration were captured at a rate of ~16 frames per second. The established CeleST software was used to process the image sequences and extract eight measures that are descriptive of *C. elegans* swim motility. As not all

images in a sequence are always successfully processed by the CeleST software, animals for which fewer than 80% of frames were valid were excluded from the analysis. For each measure, single measurements that did not fit within the range of normally observed values were deemed outliers and also excluded from the analysis. Unpaired Student's t-test was used to test for statistical significance between two strains at adult days 4 and 8.

## Life span and heat stress experiments

Life span experiments were performed as described (*Gerisch et al., 2007*). Day 0 corresponds to the L4 stage. Life spans were determined by scoring a population of 100–120 worms per genotype every day or every other day. Worms that exploded, had internal hatch, or left the plate were censored. Heat stress experiments were performed at 35°C (*Gerisch et al., 2007*). Day 1 adult worms were transferred onto pre-heated (35°C) plates 6 cm plates seeded with OP50. Worms were kept at 35°C and scored every hour for live versus dead. Experiments were repeated at least three times. Data were analyzed with Microsoft Excel and GraphPad Prism Software. p values were calculated using the Log-rank (Mantel–Cox) test to compare two independent populations.

## RNAi treatment

Worms were grown from L4 onwards unless mentioned otherwise on *E. coli* HT115 (DE3) bacteria expressing dsRNA of the target gene under the control of an IPTG-inducible promoter. RNAi colonies were grown overnight at 37°C in Luria Broth with 50 µg/ml ampicillin and 10 µg/ml tetracycline. The cultures were spun down at 4000 rpm at 4°C for 10 min. 500 µl of onefold concentrated culture was seeded onto NGM plates containing 1 M isopropyl β-D-1-thiogalactopyranoside (IPTG) to induce dsRNA expression. RNAi clones were selected from the Ahringer or Vidal library (*Kamath and Ahringer, 2003*; *Rual et al., 2004*). Clone identity was confirmed by sequencing.

## RNAi screen for *mir-1* suppressors

*mir-1(gk276)*;Q35 worms were grown from L4 on RNAi of the candidates identified in the bioinformatic or proteomic screen. Worms were maintained in the presence of FUDR (40 µM 5-fluoro-2′-deoxyuridine, Sigma) until day 8 of adulthood. The circle test was performed as described above. The number of worms that left the circle was determined after 1 min. Selected candidates identified in the circle test were examined in the body bending assay (see above). The effect of selected candidate RNAi clones was additionally tested on polyQ35 in the WT background as counter screen. For this experiment, worms were grown until day 5 of adulthood. Motility assays were performed on day 5 of adulthood because Q35 WT worms were less motile than the *mir-1(gk276)*;Q35 and were largely paralyzed by day 8.

## RNA extraction and real-time qPCR analysis

Worm populations (ca. 500 animals) were harvested on day 4 of adulthood and washed twice in cold M9. Worm pellets were taken up in 700 µl QIAzol reagent (Qiagen) and snap frozen in liquid nitrogen. Samples were subjected to four freeze/thaw cycles and homogenized with 1.0 mm zirconia/silica beads (Fisher Scientific) in a Tissue Lyser LT (Qiagen) for 15 min at full speed. After homogenization, 600 µl supernatant was transferred to fresh tubes and 120 µl chloroform were added to each tube. Components were mixed by vortexing and incubated for 2 min at room temperature. After 15 min centrifugation at 12,000 *g* 4°C, the aqueous phase was collected for total RNA extraction using the RNeasy or miRNeasy Mini Kit (Qiagen) according to the manufacturer's instructions. RNA quantity and quality were determined on a NanoDrop 2000c (PeqLab RRID:SCR_020309) and cDNA was prepared using the iScript cDNA Synthesis Kit (BioRad). To quantify RNA expression, we used the Power SYBR Green Master Mix (Applied Biosystems) on a ViiA 7 Real-Time PCR system (Applied Biosystems). Four technical replicates were pipetted on a 384-well plate using the JANUS automated workstation (PerkinElmer). Expression of target RNA was calculated from comparative CT values, normalized to *ama-1* or *cdc-42* as internal controls using the corresponding ViiA7 software. All unpublished primers were validated by determination of their standard curves and melting properties. For quantification of *mir-1*, we used TaqMan probes from Life Technologies/Thermo Fisher Scientific (Assay ID 000385) and normalized to expression of U18 as measured by TaqMan probes from Life Technologies/Thermo Fisher Scientific (Assay ID 001764). We used N = 4

independent biological replicates, and four technical replicates for every biological sample (for primers, refer to *Supplementary file 1h*).

## Western blot analysis

For western blot analysis, synchronized young adult or gravid day 1 adult worms were picked into Eppendorf tubes containing M9, snap frozen in liquid nitrogen, and lysed in 4× SDS sample buffer (Thermo Fisher) containing 50 mM DTT. After boiling and sonication, equal volumes were subjected to reducing SDS-PAGE and transferred to nitrocellulose membranes. The membranes were then blocked for 2 hr at room temperature in 5% milk in Tris-buffered Saline and Tween20 (TBST) and probed with the primary antibodies in TBST with 5% milk overnight at 4°C. Specific secondary antibodies (mouse or rabbit) were used at a concentration of 1:5000 in TBST with 5% milk at room temperature for 2 hr. The membranes were developed with Western Lightening Plus – Enhanced Chemiluminescence Substrate (PerkinElmer). Bands were detected on a ChemiDoc MP Imaging System (BioRad) and the intensity quantified using the corresponding Image Lab software (BioRad). Antibodies are listed in the Key resources table.

## Proteomic analysis

For sample collection and preparation, day 1 N2 and *mir-1(gk276)* worms were synchronized by egg laying, and $n \geq 5000$ worms per genotype were collected in M9. Samples were washed three times in M9 and directly frozen in liquid nitrogen and stored at –80°C. Five independent biological replicates of each genotype were collected for further analyses. For protein extraction, samples were thawed and boiled in lysis buffer (100 mM Tris-HCl, 6 M guanidinium chloride, 10 mM Tris(2-carboxyethyl)phosphine hydrochloride, 40 mM 2-chloroacetamide) for 10 min, lysed at high performance with a Bioruptor Plus sonicator (Diagenode) using 10 cycles of 30 s sonication intervals. The samples were then boiled again, centrifuged at 20,000 *g* for 20 min, and diluted 1:10 in 20 mM Tris pH 8.3/10% acetonitrile (ACN). Protein concentration was measured using BCA Protein Assays (Thermo Fisher). Samples were then digested overnight with rLys-C (Promega), the peptides were cleaned on a Supelco Visiprep SPE Vacuum Manifold (Sigma) using OASIS HLB Extraction cartridges (Waters). The columns were conditioned twice with methanol, equilibrated twice with 0.1% formic acid, loaded with the sample, washed three times with 0.1% formic acid, and the peptides eluted with 60% ACN/0.1% formic acid. The samples were dried at 30°C for roughly 4 hr in a Concentrator (Eppendorf) set for volatile aqueous substances. The dried peptides were taken up in 0.1% formic acid and the samples were analyzed by the Max Planck Proteomic Core facility. Mass spectrometry data acquisition, computational proteomic analysis, and differential expression analysis were performed as described (*Tharyan et al., 2020*). Upon inspection of the numbers of quantified proteins and the raw proteomic data, two replicates of N2 were excluded from further analysis. Proteomics data was deposited to the ProteomeXchange Consortium via the PRIDE partner repository (*Perez-Riverol et al., 2019*).

## Microscopy and expression analysis

For confocal microscopy, *vha-13*, *vha-5*, and *hlh-30* endogenously tagged with mNeonGreen were synchronized via egg and maintained at 25°C to induce mild stress. *hlh-30* worms were imaged as L4s and day 4 adults, while *vha-13* and *vha-5* worms as L4s, because of high mNeonGreen background expression at day 4 adulthood. Worms were anesthetized with 40 µM sodium azide, mounted on slide with 2% agar pad, and imaged with a Leica TCS SP8 microscope equipped with HC PL APO CS2 63×/1.40 oil and white light laser. Images were analyzed using Photoshop or Fiji software. To quantitate acidic lysosomes, worms grown at 25°C were incubated 48 hr prior to imaging with 2 µM LysoTracker Red DND-99 (Life Technologies). Worms were imaged as day 4 adults as described above. The number of puncta in a predefined squared area in the intestine (approximately spanning the second to fourth gut cell) was counted.

## Statistical analysis

The statistical tests performed in this study are indicated in the figure legends and in the Materials and methods. Data are represented as mean ± SEM or as individual data points, as stated

in the figure legends. The number of replicates and animals for each experiment is enclosed in their respective figure legends.

## Acknowledgements

AA would like to thank the MPI-AGE proteomics and imaging cores for services, the Caenorhabditis Genetics Center (CGC, University of Minnesota) for worm strains, the Bundesministerium für Bildung und Forschung for Sybacol funding, the Deutscher Akademischer Austauschdienst for funding, and the Max Planck Gesellschaft for core institutional support. MAM would like to thank Fundação de Amparo à Pesquisa do Estado de São Paulo (FAPESP) (grant number 2019/25958-9) and Coordenação de Aperfeiçoamento de Pessoal de Nível Superior CAPES (grant number 88881.143924/2017-01) for funding.

## Additional information

### Funding

| Funder | Grant reference number | Author |
|---|---|---|
| Bundesministerium für Bildung und Forschung | 0315893A | Adam Antebi |
| Deutscher Akademischer Austauschdienst | 57518567 | Adam Antebi |
| Max-Planck-Gesellschaft | | Adam Antebi |
| Fundação de Amparo à Pesquisa do Estado de São Paulo | 2019/25958-9 | Marcelo A Mori |
| Coordenação de Aperfeiçoamento de Pessoal de Nível Superior | 88881.143924/2017-01 | Marcelo A Mori |

The funders had no role in study design, data collection and interpretation, or the decision to submit the work for publication.

### Author contributions

Isabelle Schiffer, Conceptualization, Data curation, Formal analysis, Investigation, Visualization, Writing - original draft, Writing - review and editing; Birgit Gerisch, Conceptualization, Data curation, Formal analysis, Validation, Investigation, Visualization, Writing - original draft, Writing - review and editing; Kazuto Kawamura, Formal analysis, Validation, Investigation, Visualization, Writing - review and editing; Raymond Laboy, Formal analysis, Investigation, Visualization; Jennifer Hewitt, Formal analysis, Investigation, Visualization, Writing - review and editing; Martin Sebastian Denzel, Formal analysis, Investigation; Marcelo A Mori, Conceptualization, Writing - review and editing; Siva Vanapalli, Formal analysis, Writing - review and editing; Yidong Shen, Investigation; Orsolya Symmons, Data curation, Formal analysis, Visualization, Writing - original draft, Writing - review and editing; Adam Antebi, Conceptualization, Resources, Formal analysis, Supervision, Funding acquisition, Investigation, Writing - original draft, Project administration, Writing - review and editing

### Author ORCIDs

Kazuto Kawamura (ID) https://orcid.org/0000-0002-0376-7739
Raymond Laboy (ID) https://orcid.org/0000-0002-0375-1550
Jennifer Hewitt (ID) http://orcid.org/0000-0002-8798-1811
Martin Sebastian Denzel (ID) http://orcid.org/0000-0002-5691-3349
Orsolya Symmons (ID) http://orcid.org/0000-0003-2435-4236
Adam Antebi (ID) https://orcid.org/0000-0002-7241-3029

### Decision letter and Author response

Decision letter https://doi.org/10.7554/eLife.66768.sa1
Author response https://doi.org/10.7554/eLife.66768.sa2

# Additional files

## Supplementary files

• Supplementary file 1. Raw data. (**a**) Life span and heat stress data. (**b**) Aggregate and behavior data. (**c**) CeleST data. (**d**) List of candidate genes with in silico predicted miR-1 binding sites using microRNA.org, TargetScanWorm, and PicTar. (**e**) RNAi motility screen. (**f**) Proteomic analysis of differentially regulated proteins in *mir-1* vs. wild-type. (**g**) Microscopy data. (**h**) Primers.

• Transparent reporting form

## Data availability

The mass spectrometry proteomics data have been deposited to the ProteomeXchange Consortium via the PRIDE partner repository (Perez-Riverol et al., 2019) with the dataset identifier PXD023544. All raw data underlying the figures are provided in the supplementary file.

The following dataset was generated:

| Author(s) | Year | Dataset title | Dataset URL | Database and Identifier |
|---|---|---|---|---|
| Schiffer I | 2021 | Data from: miR-1 coordinately regulates lysosomal v-ATPase and biogenesis to impact proteotoxicity and muscle function during aging | http://proteomecentral. proteomexchange.org/ cgi/GetDataset?ID= PXD023544 | ProteomeXchange, PXD023544 |

The following previously published datasets were used:

| Author(s) | Year | Dataset title | Dataset URL | Database and Identifier |
|---|---|---|---|---|
| Boettger T | 2018 | Metabolic maturation during muscle stem cell differentiation is achieved by miR-1/133a-mediated inhibition of the Dlk-Dio3 mega gene cluster | https://www.ebi.ac.uk/ar-rayexpress/experiments/ E-MTAB-5936/?array=A-GEOD-17408 | ArrayExpress, GEOD-1740 8 |
| Zhao Y | 2014 | Identification of dysregulated genes after deletion of miR-1-1 and miR-1-2 | https://www.ncbi.nlm. nih.gov/geo/query/acc. cgi?acc=GSE45760 | NCBI Gene Expression Omnibus, GSE45760 |

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
