## [Decision Letter]

**Acceptance summary:**

This paper elegantly demonstrates how the loss of a single micro-RNA contributes to maintaining muscle function during aging under stressful conditions. Through this principal finding and their additional mechanistic results, this work suggests that the role of small non-coding micro-RNA molecules is more complex than previously understood, and will be the subject of additional future studies.

**Decision letter after peer review:**

Thank you for submitting your article "miR-1 coordinately regulates lysosomal v-ATPase and biogenesis to impact proteotoxicity and muscle function during aging" for consideration by *eLife*. Your article has been reviewed by 2 peer reviewers, and the evaluation has been overseen by Scott Leiser as Reviewing Editor and Matt Kaeberlein as the Senior Editor. The following individual involved in review of your submission has agreed to reveal their identity: Seung-Jae V Lee (Reviewer #1).

Summary of review and essential revisions:

Summary of manuscript: In this manuscript entitled "miR-1 coordinately regulates lysosomal v-ATPase and biogenesis to impact proteotoxicity and muscle function during aging", the authors demonstrated that mir-1 loss of function mutations contributes to maintaining muscle functions during aging under proteotoxic conditions in the nematode *C. elegans*. They found that deletion of mir-1 improved muscle functions in *C. elegans* that expressed polyQ, a model of Huntington's disease. Using bioinformatic and proteomic analyses, the authors identified vacuolar ATPase subunits as targets of miR-1. They then focused their work on vha-13, which encodes ATP6VIA subunit of the vacuolar ATPase, and demonstrated that miR-1 directly bound the 3' UTR of vha-13 for downregulation, which contributes to ameliorating proteotoxicity and delaying muscle aging. They also showed that miR-1 regulates lysosomal biogenesis and acidification through HLH-30/TFEB. The topic and content of this paper is novel and important, because this is the first paper demonstrating important roles of evolutionarily conserved miR-1 in muscle aging and proteostasis.

Summary of discussion: The reviewers and I are in agreement that the data of the manuscript are of extremely high quality with molecular genetics, cell biology, bioinformatics and proteomic analysis. We believe this is a worthwhile manuscript for publication in *eLife*, but should be revised with some additional data. Of highest importance, we all agree that using at least one other aggregation model (probably Q40, although others would be interesting as well) is needed, seeing as though your data disagree with a previous publication and you have the ability to test the robustness of the phenotype in multiple aggregation models. Besides this, there are a number of issues brought up during review that should be addressed in a revised manuscript, but that we believe can be addressed within the text.

1) I think the authors' efforts to identify phenotypes caused by mir-1 mutations are rigorous and comprehensive. Nevertheless, I believe it will be even better if they test one or more proteotoxic stress conditions to determine whether miR-1 mutant phenotype is specific to Q35 or general for some other proteotoxic stress conditions. For example, they can combine mir-1 mutation with other established muscle proteotoxicity models such as amyloid β or sod1 transgene in muscles. Alternatively, I wonder whether miR-1 mutations suppress phenotypes caused by hsf-1 RNAi or by ts mutations that Morimoto group used (Gidalevitz et al., 2006, Science), which may act as sensitized conditions for detecting mir-1 mutant phenotypes.

1b) It is greatly appreciated that the authors bring up the conflicting findings with the Q40 reporter versus their Q35 study. The authors should, at the very least, test the most robust assay in their arsenal with the Q40 strain. It is very possible that five

extra repeats could be causal so this would be interesting.

2) The data in the current Figure 4 are very interesting but are not as strong as the current Figure 3. Therefore, the current structure of the paper makes readers feel the paper is rather incomplete. I suggest that the authors consider changing the order of figure 3 and 4 (describing HLH-30 first, and then ATPases and VHA-13), and finishing this paper with stronger data in Figure 3.

3) The abstract suggests the authors will uncover a role distinct from development, but really the authors manipulate mIR-1 throughout life. This isn't an issue per se as the targets are manipulated post-developmentally. That said, the authors should describe the age-related expression of mIR-1 over the lifespan. Perhaps L4 (end of development), Day 1 early reproduction, Day 5 end of reproduction, and day 12 for midlife? This is important as previous work suggests that miR-1 expression is mostly developmental.

4) The mutations in putative mIR-1 sites in the vha gene family is nice, but to show sufficiency the authors should add the 3'UTR elements (single or multi) to a non-regulated gene (add to the unc-54 UTR for the GFP reporter in 3F) and show that the sequence can make the target now sensitive to mIR-1 expression.

5) The short section on HLH-30 is confusing. How is mIR-1 regulating protein localization, but not expression. This could be interesting, but as written it is out of place and detract from the detailed work on vha-13 preceding it.

*Reviewer #1:*

Aging is associated with declines in functions of almost all the tissues and organs, including muscle. miR-1, a small RNA called microRNA, functions in muscles, but the role of miR-1 in aging-associated processes remains unknown. In this paper, the authors convincingly showed that miR-1 plays protective roles in muscle functions during aging and age-associated diseases using the roundworm *Caenorhabditis elegans*. miR-1 is an evolutionarily conserved microRNA that functions in muscles in various species, including mammals. Thus, this study is very valuable for devising strategy for treating muscle aging diseases such as sarcopenia and age-associated diseases caused by proteotoxicity such as Huntington's disease.*Reviewer #2:*

In this study the authors are trying to elucidate a role for the microRNA mIR-1 in aging, which would expand upon its established roles in muscle development. The authors use a combination of powerful molecular and genetic approaches to define new targets of mIR-1that influence muscle health. The authors perform a well designed study and uncover new way to study muscle health with age. The authors conclusions are supported by the data presented and this study defines a new layer of regulation of muscle health over the lifespan, which based on the evolutionary conservation of the players has a high likelihood of informing human health.

---

## [Author Response]

Summary of review and essential revisions:Summary of manuscript: In this manuscript entitled "miR-1 coordinately regulates lysosomal v-ATPase and biogenesis to impact proteotoxicity and muscle function during aging", the authors demonstrated that mir-1 loss of function mutations contributes to maintaining muscle functions during aging under proteotoxic conditions in the nematode *C. elegans*. They found that deletion of mir-1 improved muscle functions in *C. elegans* that expressed polyQ, a model of Huntington's disease. Using bioinformatic and proteomic analyses, the authors identified vacuolar ATPase subunits as targets of miR-1. They then focused their work on vha-13, which encodes ATP6VIA subunit of the vacuolar ATPase, and demonstrated that miR-1 directly bound the 3' UTR of vha-13 for downregulation, which contributes to ameliorating proteotoxicity and delaying muscle aging. They also showed that miR-1 regulates lysosomal biogenesis and acidification through HLH-30/TFEB. The topic and content of this paper is novel and important, because this is the first paper demonstrating important roles of evolutionarily conserved miR-1 in muscle aging and proteostasis.Summary of discussion: The reviewers and I are in agreement that the data of the manuscript are of extremely high quality with molecular genetics, cell biology, bioinformatics and proteomic analysis. We believe this is a worthwhile manuscript for publication in eLife, but should be revised with some additional data. Of highest importance, we all agree that using at least one other aggregation model (probably Q40, although others would be interesting as well) is needed, seeing as though your data disagree with a previous publication and you have the ability to test the robustness of the phenotype in multiple aggregation models. Besides this, there are a number of issues brought up during review that should be addressed in a revised manuscript, but that we believe can be addressed within the text.

Thank you for this excellent suggestion. We measured protein aggregation and motility in polyQ40 and α-synuclein proteotoxicity models. We added the following text in the results:

“To determine whether *mir-1* loss generally protects against proteotoxic stress, we

examined two other muscle-specific proteotoxic models: polyQ40 and human α-synuclein (α-syn). In contrast to polyQ35, which forms aggregates during adulthood, polyQ40 aggregates were already visible during development. We found aggregate numbers significantly reduced in *mir-1(gk276)*;Q40 L4 worms (Supplemental Table 2). However, the motility of *mir-1(gk276)*;Q40 worms, determined by 1 minute thrashing assays, was unaffected at day 3 of adulthood and reduced in 2 out of 3 experiments at day 7 of adulthood compared to Q40 controls (Supplemental Table 2). Similarly aggregate numbers were reduced in *mir-1(gk276)*;α-syn worms at day 1 of adulthood compared to the α-syn control (Supplemental Table 2). Thrashing of *mir-1(gk276)*;α-syn worms was significantly reduced on day 1 of adulthood, but by day 5 there was no significant difference in motility between *mir-1(gk276)*;α-syn and α-syn worms (Supplemental Figure 1O, Supplemental Table 2). Taken together, *mir-1* loss reduced aggregate numbers in 3 different proteotoxic stress paradigms. However, reduced number of aggregates did not always directly correlate with improved motility, reinforcing the notion that protein aggregation may promote or aggravate health depending on the proteotoxic species and age (Cohen, Bieschke, Perciavalle, Kelly, and Dillin, 2006).”

In the discussion we included the following text:

“Recently Pocock and colleagues reported that miR-1 downregulates the Rab GTPase TBC-7/TBC1D15, thereby promoting autophagic flux in worms and human cells

(Nehammer et al., 2019). In *C. elegans* they observed that *mir-1(+)* ameliorates polyQ40

proteotoxicity, seemingly contradicting our results. We found *mir-1* deletion to reduce

aggregate numbers in polyQ35, polyQ40 and α-syn models, while effects on motility varied among the three models. The relationship between aggregation and cellular toxicity remains controversial, and the uncoupling of aggregation and toxicity is observed in various disease models. Mammalian studies suggest context-dependent *mir-1* effects, where *mir-1* either promotes or inhibits autophagy (Ejlerskov, Rubinsztein, and Pocock, 2020; Hua, Zhu, and Wei, 2018; Xu, Cao, Zhang, Zhang, and Yue, 2020). Whether miR-1 is proteoprotective or limits proteoprotection in *C. elegans* could hinge on many factors. Because microRNAs generally fine tune regulation, and often work in feedback circuits, their impact on physiology can be subtle, and small differences in culture conditions, genetic background, metabolic state, aggregation onset during development or ageing could give rise to divergent outcomes.”

We added the data to Supplemental Table 2.

1) I think the authors' efforts to identify phenotypes caused by mir-1 mutations are rigorous and comprehensive. Nevertheless, I believe it will be even better if they test one or more proteotoxic stress conditions to determine whether miR-1 mutant phenotype is specific to Q35 or general for some other proteotoxic stress conditions. For example, they can combine mir-1 mutation with other established muscle proteotoxicity models such as amyloid β or sod1 transgene in muscles. Alternatively, I wonder whether miR-1 mutations suppress phenotypes caused by hsf-1 RNAi or by ts mutations that Morimoto group used (Gidalevitz et al., 2006, Science), which may act as sensitized conditions for detecting mir-1 mutant phenotypes.

We tested the effect of *mir-1(gk276)* as suggested on Q40 and α-synuclein and added

the data to the paper (see above).

1b) It is greatly appreciated that the authors bring up the conflicting findings with the Q40 reporter versus their Q35 study. The authors should, at the very least, test the most robust assay in their arsenal with the Q40 strain. It is very possible that 5 extra repeats could be causal so this would be interesting.

We tested the effect of *mir-1(gk276)* on Q40 and added the data (see above).

2) The data in the current Figure 4 are very interesting but are not as strong as the current Figure 3. Therefore, the current structure of the paper makes readers feel the paper is rather incomplete. I suggest that the authors consider changing the order of figure 3 and 4 (describing HLH-30 first, and then ATPases and VHA-13), and finishing this paper with stronger data in Figure 3.

Thank you for this suggestion. We rearranged the text to mention the HLH -30 results

before VHA-13. We now say:

“We further investigated whether miR-1 regulates these gene products at the

protein level. We endogenously tagged *vha-5, vha-10, vha-13*, *vha-19* (N-terminus) as

well as *hlh-30* (C-terminus) by CRISPR/Cas9 using 3xFlag-mNeonGreen-tag. Tagging

*vha-19* and *vha-10* caused lethality and were not pursued further. Tagged *vha-5* showed

significant upregulation in the pharynx in *mir-1(gk276)* mutants compared to WT, but not in vulva muscle (Supplemental Figure 3A and B, Supplemental Table 7). However, *vha-5* RNAi knockdown had no significant difference in motility compared to controls

(Figure 2E-G, Supplemental Table S2). Further, we saw no obvious regulation of endogenously tagged *hlh-30* protein levels in the pharynx in late L4 or day 4 adults (Supplemental Figure 4A and B, Supplemental Table 7), though we did see increased nuclear localization in the hypodermis in day 4 *mir-1(gk276)* adults (Figure 4A and 4B).

N-terminally tagged *vha-13(syb586),* however, was viable, showed expression that

was regulated, and harboured two potential *mir-1* binding sites in the 3’UTR (Figure 3F,

Supplemental Figure 3C).”

Although we changed the text, we nevertheless kept HLH-30::NeonGreen data in Figure

4, along with lysosome biogenesis data. This is because we feel that swapping Figure 3

and 4 would interrupt the narrative that goes from *mir-1* target to broader lysosomal

physiology.

3) The abstract suggests the authors will uncover a role distinct from development, but really the authors manipulate mIR-1 throughout life. This isn't an issue per se as the targets are manipulated post-developmentally. That said, the authors should describe the age-related expression of mIR-1 over the lifespan. Perhaps L4 (end of development), Day 1 early reproduction, Day 5 end of reproduction, and day 12 for midlife? This is important as previous work suggests that miR-1 expression is mostly developmental.

We added data from microRNA-seq analysis of wild-type worms at day 1 of adulthood

(start of adulthood), day 7 (around the end of the reproductive period), day 14, and day

21 (around the mean lifespan of wild-type worms) (Yifei Zhou et al., 2019).

We added the following sentence in the manuscript:

“While microRNAs are known as important developmental regulators (Ivey and Srivastava, 2015), miR-1 expression also persists throughout adulthood, decliningby half from day 1 to day 14 of adulthood (Supplemental Figure 1A, data from (Zhouet al., 2019).”

4) The mutations in putative mIR-1 sites in the vha gene family is nice, but to show sufficiency the authors should add the 3'UTR elements (single or multi) to a non-regulated gene (add to the unc-54 UTR for the GFP reporter in 3F) and show that the sequence can make the target now sensitive to mIR-1 expression.

The reviewer is correct in pointing out that we did not pursue experiments to show that

miR-1 binding sites are sufficient for regulation. Our aim for this line of experiments was to show that miR-1 directly regulates *vha-13* expression, and to determine the

physiologically relevant miR-1 binding sites in the *vha-13* 3’UTR. We successfully

identified two miR-1 binding sites, which are necessary for miR-1 dependent regulation of *vha-13* expression. In theory, it is possible that introducing these two binding sites into an *unc-54* 3’UTR may confer regulation. However, past examples of microRNA regulation indicate that surrounding sequences of the binding site are also important, as well as secondary structures of the 3’ UTR, and the microRNA binding sites of *lsy-2* or *let-7* are not sufficient (Didiano and Hobert, 2008; Brancanti and Grosshans 2018). These studies actually used the *unc-54* 3’UTR. By inference, miR-1 binding sites would also likely not be sufficient.

Finally, as an example from our proteomics data, the *unc-44* 3’UTR carries the same two

miR-1 binding sites as *vha-13* (ACATTCCA and ATTCCA), but does not show the expected derepression in the miR-1 deletion strain. Given these observations, we feel

that demonstrating sufficiency would be highly unlikely and lies outside of the biologically relevant context and outside of the scope of this paper.

5) The short section on HLH-30 is confusing. How is mIR-1 regulating protein localization, but not expression. This could be interesting, but as written it is out of place and detract from the detailed work on vha-13 preceding it.

We now mention *hlh-30* briefly ahead of the *vha-13* work as in point 2, and say:

“Further, we saw no obvious regulation of endogenously tagged *hlh-30* protein levels in

the pharynx in late L4 or day 4 adults (Supplemental Figure 4A and B, Supplemental

Table 7), though we did see increased nuclear localization in the hypodermis in day 4

*mir-1(gk276)* adults (Figure 4A and 4B).”

As far as how miR-1 regulates the nuclear localization of HLH-30, we can only speculate

that it does so by affecting mTOR signaling. We now say in the discussion:

“Notably, miR-1 predicted targets also include pro-aging components of mTOR signalling such as *let-363/*mTOR and *raga-1/*RagA (targetscan.org), which also regulate

the activity or localization of these transcription factors.”